# Nicotinic acetylcholine receptor signaling maintains epithelial barrier integrity

Nadja S Katheder[1], Kristen C Browder[1], Diana Chang[2], Ann De Maziere[3], Pekka Kujala[3], Suzanne van Dijk[3], Judith Klumperman[3], Tzu-Chiao Lu[4], Hongjie Li[4,5], Zijuan Lai[6], Dewakar Sangaraju[6], Heinrich Jasper[1]*

[1]Regenerative Medicine, Genentech, South San Francisco, United States; [2]Human Genetics, Genentech, South San Francisco, United States; [3]Center for Molecular Medicine, Cell Biology, University Medical Center Utrecht, Utrecht, Netherlands; [4]Huffington Center on Aging, Baylor College of Medicine, Houston, United States; [5]Department of Molecular and Human Genetics, Baylor College of Medicine, Houston, United States; [6]Drug Metabolism and Pharmacokinetics, Genentech, South San Francisco, United States

**Abstract** Disruption of epithelial barriers is a common disease manifestation in chronic degenerative diseases of the airways, lung, and intestine. Extensive human genetic studies have identified risk loci in such diseases, including in chronic obstructive pulmonary disease (COPD) and inflammatory bowel diseases. The genes associated with these loci have not fully been determined, and functional characterization of such genes requires extensive studies in model organisms. Here, we report the results of a screen in *Drosophila melanogaster* that allowed for rapid identification, validation, and prioritization of COPD risk genes that were selected based on risk loci identified in human genome-wide association studies (GWAS). Using intestinal barrier dysfunction in flies as a readout, our results validate the impact of candidate gene perturbations on epithelial barrier function in 56% of the cases, resulting in a prioritized target gene list. We further report the functional characterization in flies of one family of these genes, encoding for nicotinic acetylcholine receptor (nAchR) subunits. We find that nAchR signaling in enterocytes of the fly gut promotes epithelial barrier function and epithelial homeostasis by regulating the production of the peritrophic matrix. Our findings identify COPD-associated genes critical for epithelial barrier maintenance, and provide insight into the role of epithelial nAchR signaling for homeostasis.

*For correspondence:
jasperh@gene.com

## Editor's evaluation

This study presents valuable findings on the maintenance of the intestinal epithelial barrier function in *Drosophila* that involves the production of the peritrophic matrix rather than the epithelial junctions, and highlights a critical role of Ach/nAchR signaling there. The methods, data, and analyses broadly support the claims in line with current state-of-the-art.

## Introduction

Barrier epithelia such as the skin, linings of the gastrointestinal and urogenital tracts, and the airways play a critical role in maintaining a strict separation of external and internal environments, yet also enable the exchange of gases, water, nutrients, and immune mediators. They serve as a first layer of defense against external insults and possess remarkable regenerative capacity that declines with age (*Jasper, 2020*). In *Drosophila*, loss of intestinal barrier function is accompanied by commensal dysbiosis and inflammation and reliably predicts impending organismal death (*Rera et al., 2012*). Similarly,

increased barrier permeability and changes in microbiome composition and abundance have been reported in various human diseases, such as inflammatory bowel disease and chronic obstructive pulmonary disease (COPD) (*Raftery et al., 2020*).

COPD is a major contributor to global morbidity and mortality and is characterized by an obstructed airflow resulting in shortness of breath upon exertion. At the tissue level, lungs of COPD patients display chronic inflammation, extensive cellular remodeling, and barrier dysfunction (*Aghapour et al., 2018*; *Barnes, 2019*; *Carlier et al., 2021*).

While smoking or exposure to environmental air pollutants remain major risk factors, many COPD patients are non-smokers, suggesting a genetic component contributing to disease susceptibility (*Aghapour et al., 2022*; *Barnes, 2019*). Several genome-wide association studies (GWAS) have been performed in which risk loci for incidence of COPD have been identified (*Hobbs et al., 2017*; *Pillai et al., 2009*; *Sakornsakolpat et al., 2019*). One of the most well-known risk loci is located near the nicotinic acetylcholine receptor (nAchR) CHRNA3/5 genes and has also been associated with increased nicotine dependence and smoking behavior, and lung cancer (*Amos et al., 2008*; *Carlier et al., 2021*; *Cui et al., 2014*; *Hobbs et al., 2017*; *Hung et al., 2008*; *Pillai et al., 2009*; *Wilk et al., 2012*). Recent work has demonstrated a role for CHRNA5 in the formation of COPD-like lesions in the respiratory epithelium independently of cigarette smoke, suggesting a direct involvement of nAchRs in shaping epithelial integrity (*Routhier et al., 2021*). The endogenous ligand of nAchR, acetylcholine (Ach), is a classic neurotransmitter synthesized by choline acetyltransferase (ChAT) in cholinergic neurons, as well as in immune cells and epithelial cells, such as brush/tuft cells (*Kummer and Krasteva-Christ, 2014*; *Wessler and Kirkpatrick, 2008*). Such cells orchestrate type 2 inflammatory responses (*Parker et al., 2019*; *Sell et al., 2021*), mucociliary clearance (*Perniss et al., 2020*), and limit biliary inflammation (*O'Leary et al., 2022*; *O'Leary et al., 2019*). How Ach influences homeostasis of barrier epithelia and how disease-associated nAchR variants perturb epithelial function remain mostly unclear.

Overall, experimental evidence for the involvement of specific genes associated with the COPD risk loci identified in these studies is mostly lacking, and will be essential for the development of therapeutic strategies targeting novel pathways. The use of genetically accessible model systems with enough physiological complexity to model cell and tissue interactions in barrier epithelia may help accelerate the evaluation of potential disease-causing genes identified in COPD GWAS. To test this idea, we have here used the *Drosophila* midgut as a genetically accessible model for epithelial barrier homeostasis to interrogate genes predicted to be involved in COPD based on GWAS. The *Drosophila* intestine is lined by a pseudostratified epithelium consisting of enterocytes (ECs) and enteroendocrine cells (EEs) that are regenerated from a basal population of intestinal stem cells (ISCs) (*Miguel-Aliaga et al., 2018*). In its structure, cell composition and molecular regulation of regenerative processes, the fly intestinal epithelium resembles mammalian airway epithelia (*Biteau et al., 2011*).

Under stress conditions, in response to enteropathogen infection, as well as during normal aging, the fly intestinal epithelium loses its barrier function and exhibits stem cell hyperplasia and commensal dysbiosis (*Jasper, 2020*). These phenotypes recapitulate changes observed in airway epithelia of COPD patients and can thus be used as a model for pathophysiological changes occurring in this disease (*Carlier et al., 2021*; *Raftery et al., 2020*).

To assess the role of candidate genes associated with risk alleles in COPD GWAS in the maintenance of barrier epithelia integrity, we performed an RNA interference (RNAi) screen perturbing their *Drosophila* orthologs systemically and quantifying the impact of these perturbations on intestinal barrier function. Several of the candidate genes identified in this screen as required for barrier integrity encode for subunits of the nAchR.

In the fly intestine, we find that ChAT is expressed by a subset of EEs and that EC-specific expression of nAchR is required for barrier integrity by stimulating chitin release and ensuring maintenance of the peritrophic matrix (PM), a chitinous structure protecting the epithelium from luminal insults. In ECs, Ach is required for the expression of Syt4, a critical regulator of exocytosis (*Yoshihara et al., 2005*; *Zhang et al., 2011*) which is required for the maintenance of PM structure and epithelial barrier function. Our data illustrate the usefulness of *Drosophila* as a model for prioritization of potential disease genes identified in GWAS, and identify nAchR signaling as a critical mediator of epithelial homeostasis in barrier epithelia.

## Results

### A genetic screen assessing the role of COPD candidate genes in barrier function

To obtain a curated candidate gene list for COPD, we assigned candidate genes to COPD risk loci (*Hobbs et al., 2017*) using a combination of expression quantitative trait loci (eQTL), coding annotation, and distance-based metrics (see Materials and methods, *Supplementary file 1*). *Drosophila* orthologs were identified with the DRSC integrative ortholog prediction tool (DIOPT, *Hu et al., 2011*) and corresponding hits with the highest DIOPT score were selected, resulting in a total of 33 *Drosophila* genes screened initially (*Figure 1A*).

We perturbed these genes systemically by RNAi in an inducible fashion using the ubiquitous RU486-inducible Gal4 driver da-GeneSwitch (da-GS) and scored epithelial barrier dysfunction in homeostatic and stress conditions using the 'smurf assay' (*Rera et al., 2011*). In this approach flies are fed food containing a non-absorbable blue food dye. If the intestinal epithelial barrier is compromised, the dye leaks into the open circulatory system and gives the fly a blue appearance reminiscent of the popular blue cartoon characters. Where available, a minimum of two different RNAi lines per gene were included (*Supplementary file 2*). Female flies carrying the da-GS driver and RNAi construct were allowed to mature and mate for 10–12 days before being placed on blue food with RU486 to induce knockdown for 24 hr. Since COPD is strongly associated with environmental stress, we then challenged flies with Paraquat (*N*,*N*'-dimethyl-4,4'-bipyridinium dichloride), a herbicide known to inflict oxidative stress and damage to the fly gut comparable to the effects of cigarette smoke on the lung epithelium (*Biteau et al., 2008*; *Caliri et al., 2021*). After 16 hr Paraquat challenge the flies were moved back to blue food containing RU486 and smurf numbers were recorded over the span of about a week (*Figure 1B*). We generated a 'barrier dysfunction index' for every RNAi line by calculating the natural logarithm (ln) of the ratio of peak smurf percentage between RNAi line and control knockdown and plotted individual RNAi lines accordingly. A positive index implies an enhancement of barrier dysfunction after depletion, while a negative index suggests rescue of barrier integrity after depletion (*Figure 1C*). Based on the outcomes of individual RNAi knockdowns, we assigned an overall rating for each candidate gene (*Supplementary file 2*). We found that disruption of 17 genes (~52%) resulted in enhancement (e.g. these genes were necessary for barrier integrity), while disruption of 4 genes (12%) resulted in suppression of the barrier dysfunction. The remaining 12 genes did not display any effect on barrier function (*Figure 1D*). Out of the 16 of *Drosophila* hits where eQTL data was available for the corresponding human gene, 9 were consistent with the direction of the effect inferred from the association of the COPD risk allele with gene expression (56%, *Figure 1A*, *Supplementary file 2*).

### nAchR subunit expression in ECs is required for barrier function

Our initial screen identified disruption of five nAchR subunits as a strong enhancers of barrier dysfunction. Ubiquitous knockdown of various nAchR subunits with da-GS lead to mild barrier dysfunction under homeostatic conditions, and greatly enhanced barrier dysfunction after Paraquat challenge (*Figure 2A*, *Figure 2—figure supplement 1B*), suggesting a sensitization of the epithelium to stress. To account for different genetic backgrounds of RNAi lines, we tested a range of control lines and did not observe any significant differences between them (*Figure 2—figure supplement 1A*).

Ach is the physiological ligand for nAchRs and is produced by ChAT, an enzyme that catalyzes the transfer of an acetyl group from coenzyme acetyl-CoA to choline (*Taylor and Brown, 1999*). Modulation of total organismal Ach levels by RNAi-mediated silencing of ChAT under control of da-GS also resulted in increased barrier dysfunction after Paraquat exposure (*Figure 2B*, *Figure 2—figure supplement 1C*), further supporting the role of Ach/nAchR signaling in maintaining intestinal epithelial homeostasis.

To investigate a possible direct intestinal role for nAchR, and to identify the requirement for individual subunits, we used the drivers NP1-Gal4 and 5966-GS to separately deplete nAchR subunits. The former induces expression of UAS-linked transgenes in ECs, while the latter targets enteroblasts and ECs (*Jiang et al., 2009*; *Zeng and Hou, 2015*). While 5966-GS is inducible using RU486, we combined NP1-Gal4 with tubG80[ts] (NP1[ts]) to allow for temperature-mediated induction (TARGET system; *McGuire et al., 2004*) before subjecting the flies to Paraquat. Knockdown of nAchR α4 or β3 with both drivers increased the numbers of smurf flies, indicating a defective epithelial barrier (*Figure 2C and D*, *Figure 2—figure supplement 1D, E*).

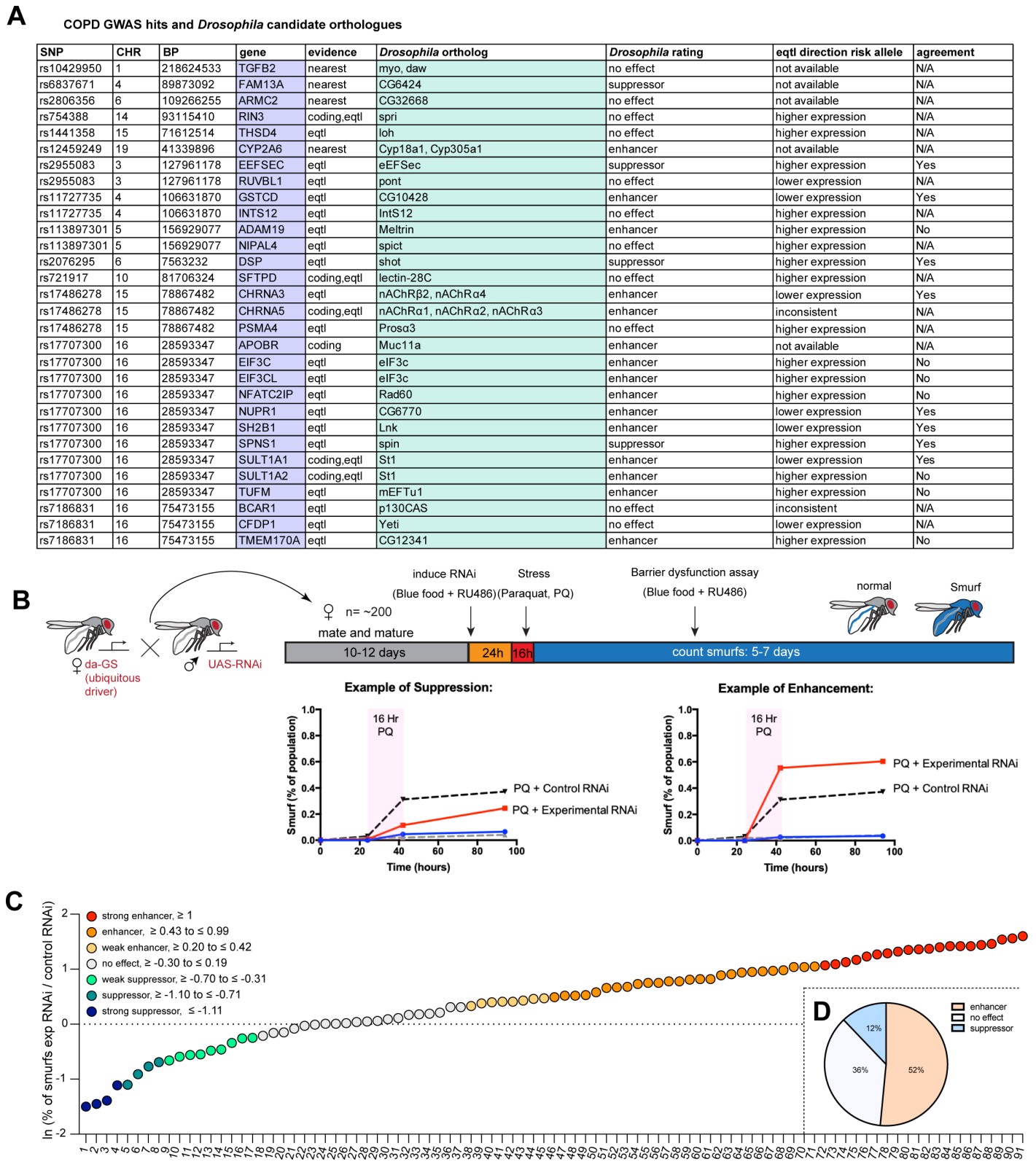

**Figure 1.** A *Drosophila* screen for chronic obstructive pulmonary disease (COPD)-associated candidate genes. (**A**) List of human candidate genes for genetic loci associated with COPD risk and their *Drosophila* orthologs. An overall rating was assigned to the *Drosophila* genes based on the detailed results of the individual RNA interference (RNAi) lines included in the screen: Genes exacerbating barrier dysfunction upon depletion were categorized as enhancers, while genes whose depletion improved barrier function were rated as suppressors of barrier dysfunction (***Supplementary***

*Figure 1 continued on next page*

*Figure 1 continued*

*file 2*). When available, the human risk allele expression data is compared to the results of the *Drosophila* screen (agreement column) (SNP, single nucleotide polymorphism; CHR, chromosome; BP, base pair number; eQTL, expression quantitative trait loci). (**B**) Experimental design of intestinal barrier function screen. Flies carrying the ubiquitous driver da-GeneSwitch (da-GS) were crossed to RNAi lines targeting candidate genes. The female offspring were aged for 10–12 days before induction of RNAi expression by RU486 for 24 hr on blue food. Flies were challenged with sucrose alone (mock) or 25 mM Paraquat (PQ) for 16 hr overnight and then placed back on blue food with RU486. Blue flies with a defective intestinal barrier ('smurfs') were counted daily for 5–7 days. (**C**) Ranking of screened RNAi lines based on the natural logarithm (ln) of the ratio between the proportion of smurfs after candidate gene knockdown and Luciferase RNAi control. Each number corresponds to an RNAi line listed in *Supplementary file 2*. Cutoffs for the different categories are indicated. (**D**) Summary of screen results based on broad categorization as enhancer, suppressor, or no effect. If several RNAi lines targeting the same gene unanimously had no effect, the gene was rated 'no effect, conclusive', while inconsistent results were rated 'no effect, inconclusive'. For details, see *Supplementary file 2*.

The online version of this article includes the following source data for figure 1:

**Source data 1.** Editable table of *Figure 1A*.

Knockdown of nAchR in ECs resulted in various hallmarks of epithelial stress. These include induction of ISC proliferation (*Figure 2E*), presumably due to stress signals released by ECs (*Biteau et al., 2011*), as well as activation of JAK/STAT signaling (measured using the 2xSTAT::GFP reporter; *Bach et al., 2007*) and ER stress signaling (measured using an Xbp1-eGFP reporter; *Sone et al., 2013*; *Figure 2F and G*). Interestingly, organization of epithelial junctions in ECs remained mostly unaltered, as visualized by staining for the septate junction marker Dlg and Coracle as well as localization of arm-GFP, suggesting that barrier dysfunction may be caused by a separate mechanism (*Figure 2H and I*, *Figure 2—figure supplement 2A*).

To further confirm and characterize the role for nAchR subunits in epithelial homeostasis, we specifically depleted nAchR subunits in stem cells using esg-Gal4, UAS-2xEYFP; Su(H)GBE-Gal80, tub-Gal80$^{ts}$ (ISC$^{ts}$). Individual knockdown of nAchR α2, α4, β1, or β3 resulted in increased barrier dysfunction after Paraquat challenge as well as decreased survival after *P. entomophila* (PE) infection (*Figure 2—figure supplement 2B, C*). When challenged with PE, stem cells depleted for various nAchR subunits underwent mitosis at a similar rate as their control counterparts (*Figure 2—figure supplement 2D*), suggesting that barrier dysfunction is not caused by an inability of stem cells to regenerate the epithelium. We generated MARCM mutant clones (*Lee and Luo, 2001*) lacking *nAChRα2* using the null allele *nAChRα2$^{attP}$* generated by CRISPR/Cas9-based homologous recombination resulting in the introduction of an attP site, 3xP3-RFP, and a loxP site (*Deng et al., 2019*; *Lu et al., 2022*). Clone formation, growth, and cell composition also provide insight into a possible role of *nAChRα2* in ISC proliferation and differentiation. Consistent with the results of the ISC-specific knockdown, *nAChR α2$^{attP}$* clones grew to similar cell numbers as their control counterparts. Interestingly, however, they failed to produce normal numbers of EEs, as only 32% of *nAChR α2* clones contained at least 1 EE compared to 72% of clones in the control samples (*Figure 2J*). Similar results (although the reduction of EE numbers was not significant) were also observed in loss of function MARCM clones for a separate subunit, *nAChR α1* (*Figure 2—figure supplement 2E*). In addition to a broader role for *nAchR* in maintaining barrier integrity, *nAchR* may thus also be required for the proper differentiation of EEs. We also knocked down additional subunits, generating clones depleted for these subunits via RNAi using the esgF/O approach (*Jiang et al., 2009*). Depletion of nAchR α4 and β3 slightly reduced overall size of the clones compared to control, and we detected a non-significant trend toward fewer EEs after nAchR depletion (*Figure 2—figure supplement 2F*, the weak phenotypes observed in these RNAi experiments may be a consequence of incomplete knockdown efficiency).

## Ach promotes barrier function

We sought to identify the source of Ach activating these receptors in the gut epithelium next. Because of its well understood role as a neurotransmitter, we initially focused on the innervation of the fly gut, which has been described previously (*Cognigni et al., 2011*). Expression of UAS-GFP under the control of two independently established ChAT-Gal4 drivers confirmed that some of these neurons are indeed cholinergic (*Figure 3A and B'*, *Figure 3—figure supplement 1A*). Upon closer examination of the epithelium, we also noticed a subset of prospero-positive EEs expressing GFP, predominantly located in the R4 and R5 regions of the midgut (*Figure 3B and C*, *Figure 3—figure supplement 1A*). In addition, labeling of guts expressing GFP under the control of prospero-Gal4 or ChAT-Gal4 combined

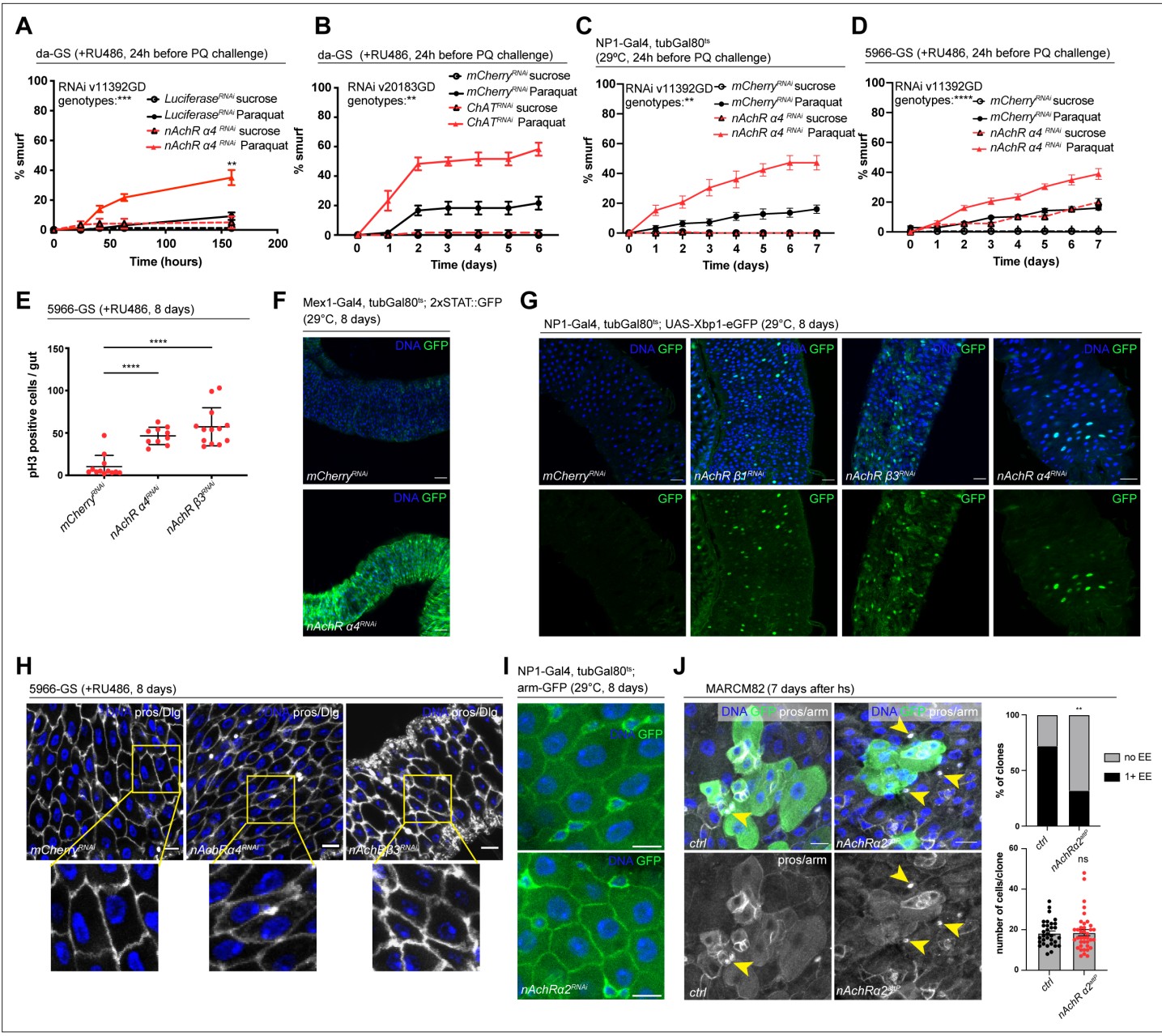

**Figure 2.** Nicotinic acetylcholine receptor (nAChR) genes are required for barrier function in enterocytes (ECs) and enteroendocrine (EEs) cell differentiation. (**A**) Barrier dysfunction assay after Luciferase (control) or nAChR α4 subunit depletion for 24 hr with ubiquitous driver da-GeneSwitch (da-GS). nAChR α4: n=100 for Luciferase RNAi (control) on sucrose; n=125 animals for Luciferase RNAi on sucrose + Paraquat; n=150 for nAChR α4 RNAi on sucrose; n=175 animals for nAChR α4 on sucrose + Paraquat. Paraquat concentration 25 mM. N=1. Two-way repeated measures (RM) ANOVA (significance stated next to 'genotypes'). (**B**) Barrier dysfunction assay after mCherry (control) or choline acetyltransferase (ChAT) depletion for 24 hr with ubiquitous driver da-GS. n=75 animals per genotype and condition; N=3. Paraquat concentration 15 mM. Two-way RM ANOVA. (**C**) Barrier dysfunction assay after mCherry (control) or nAChR α4 depletion for 24 hr with EC-specific driver NP1-Gal4, tubGal80ts (NP1ts). n=125 animals per genotype and condition; N=3. Two-way RM ANOVA. (**D**) Barrier dysfunction assay after mCherry (control) or nAChR α4 depletion for 24 hr with EC and enteroblast-specific driver 5966-GS. n=175 animals per genotype and condition; N=3. Two-way RM ANOVA. (**E**) Quantification of intestinal stem cell (ISC) mitoses in guts depleted of nAChR β3 and α4 subunits in ECs for 8 days. Mitotically active ISCs are labeled with anti-pH3 antibody; n=12;10;13 guts for mCherry (control), nAChR α4 RNAi and nAChR β1 RNAi, respectively. N=3. Ordinary one-way ANOVA followed by Dunnett's multiple comparisons test. (**F**) Confocal microscopy images of guts expressing a 2xSTAT::GFP reporter (green) depleted of mCherry (control) or nAChR α4 for 8 days in ECs with Mex-Gal4, tubGal80ts. n=10 guts per genotype. N=3. Scale bar 50 µm. (**G**) Confocal immunofluorescence image of posterior midguts expressing the UPR-reporter UAS-Xbp1-eGFP (green) after 8 days of nAChR subunit knockdown by RNAi. The EGFP tag is only in frame with the Xbp1(s) coding sequence after splicing using the unconventional splice site, which occurs under stress conditions. DNA (blue) is labeled with Hoechst. n=8 guts per

*Figure 2 continued on next page*

*Figure 2 continued*

genotype. N=3. Scale bar 25 μm. (**H**) Confocal immunofluorescence images examining epithelial organization of septate junctions stained with anti-Dlg antibody (white), DNA (blue) is labeled with Hoechst. Yellow boxed insets are shown enlarged in bottom row. n=8 guts per genotype. N=3. Scale bars 10 μm. (**I**) Confocal microscopy images of guts expressing GFP-tagged armadillo (arm-GFP, green) depleted of mCherry (control) or nAchR α4 for 8 days in ECs with NP1-Gal4, tubGal80ᵗˢ. n=8 guts per genotype. N=3. Scale bar 10 μm. (**J**) Confocal immunofluorescence images of wildtype and nAchR α2 MARCM clones (green) 7 days after heat shock. Stem cells and enteroblasts are stained with anti-armadillo antibody (white); EEs are labeled with anti-prospero antibody (white, nuclear signal highlighted with yellow arrowheads) and DNA (blue) is labeled with Hoechst. Scale bars 15 μm. Quantification of EE numbers within clones: n=32;38 clones for wildtype or nAchR α2, respectively, from three pooled experiments. Fisher's exact test. Quantification of cell numbers/clone: n=32;38 clones for wildtype or nAchR α2, respectively, from three pooled experiments. Unpaired two-tailed t-test. Data presented as mean ± SEM. ns, not significant, p>0.05; *p≤0.05; **p≤0.01; ***p≤0.001; ****p≤0.0001. n: number of animals or midguts analyzed; N: number of independent experiments performed with similar results and a similar n.

The online version of this article includes the following figure supplement(s) for figure 2:

**Figure supplement 1.** Ubiquitous nicotinic acetylcholine receptor (nAchR) signaling is required for barrier function.

**Figure supplement 2.** nAchR signaling in ISCs and ECs promotes barrier function independently of junctional integrity.

with tubGal80ᵗˢ (prosᵗˢ and ChATᵗˢ) with a ChAT antibody confirmed ChAT expression in a subset of EEs (*Figure 3D*, *Figure 3—figure supplement 1B*). ChAT antibody staining labeled more cells than ChAT-Gal4 (Mi{Trojan-GAL4.0}ChAT[MI04508-TG4.0] CG7715[MI04508-TG4.0-X]), suggesting that this driver may not fully capture all EEs expressing ChAT. Depletion of nAchR subunits in ECs did not affect the number of EEs (*Figure 3—figure supplement 1C*).

To address the role of Ach production in barrier integrity, we depleted ChAT with prosᵗˢ, as well as with ChATᵗˢ. Reduction of ChAT levels with both drivers rendered flies more susceptible to barrier dysfunction after Paraquat exposure (*Figure 3E and F*). Prospero is a known neuronal driver (*Balakireva et al., 1998*) and is also expressed in enteric neurons (*Figure 3—figure supplement 1D*). We screened additional drivers in order to separate the neuronal and epithelial contribution of Ach to barrier function, such as CG32547-Gal4 (*Guo et al., 2019*), which also presented expression in enteric neurons (*Figure 3—figure supplement 2A*) and Orcokinin-Gal, a driver identified through the publicly available scRNA data from the fly cell atlas (*Li et al., 2022*). While Orcokinin transcript levels were low in body neurons and high in EEs, this driver is also partially expressed in ECs, thus preventing a clean separation of Ach sources and their contributions to barrier function (*Figure 3—figure supplement 2A*). Further attempts to leverage previously identified neuropeptide drivers or a split-Gal4 approach did not yield a clean separation of neuronal and EE labeling, either (*Figure 3—figure supplements 2 and 3*).

Combined, these data support the notion that Ach signaling is critical to maintain barrier integrity and stress resilience in the intestinal epithelium of the fly. While cholinergic innervation is a likely source of the ligand in this response, local production of Ach by EEs may also play a role in maintaining homeostasis.

## Transcriptional changes after disruption of Ach signaling in the intestinal epithelium

As we observed barrier dysfunction without obvious deregulation of epithelial junctions after nAChR loss in ECs, we decided to profile changes in gene expression elicited in the gut by nAchR depletion. We performed RNAseq on whole guts depleted for nAchR β1 or β3 for 3 days under the control of NP1ᵗˢ. PCA suggested that the transcriptomes from intestines with nAchR knockdown were clearly distinct from transcriptomes of intestines with a control RNAi construct (mCherry RNAi; *Figure 4A*). Overall, we observed 240 upregulated and 215 downregulated genes (*Figure 4B*; FDR ≤ 0.1; log2(-fold change) <–1 or >1; 100% of samples have ≥1 reads), of which 171 were differentially expressed in both nAchR β1 and β3 knockdowns, supporting the idea that these subunits have partially overlapping functions (*Figure 4B*, *Figure 4—figure supplement 1A, F*). Synaptotagmin 4 (Syt4) was the most significantly downregulated gene in both knockdowns (*Figure 4D*). RT-qPCR analysis confirmed a reduction of Syt4 levels after nAchR depletion (*Figure 4—figure supplement 1B*). Gene ontology (GO) term enrichment analysis revealed increased expression of glucosidases and hydrolases after nAchR knockdown (*Figure 4—figure supplement 1C, F*), and downregulation of genes involved in immune responses such as lysozymes (*Figure 4C*, *Figure 4—figure supplement 1D*) and genes related to chitin binding and metabolism (*Figure 4C*).

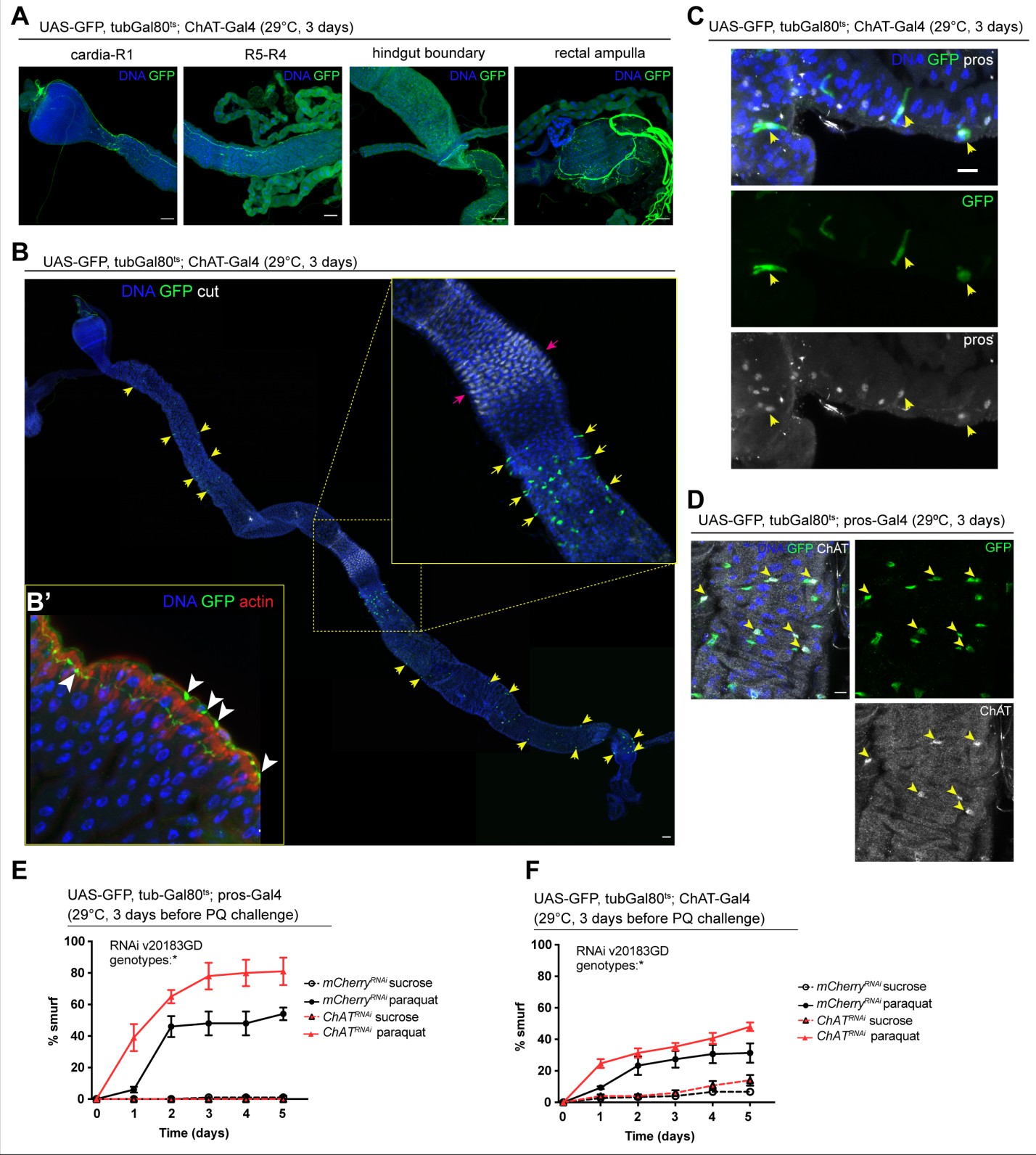

**Figure 3.** Acetylcholine produced in enteroendocrine cells (EEs) and/or neurons promotes barrier function. (**A**) Confocal immunofluorescence image of cholinergic innervation of different intestinal compartments. GFP (green) expression is driven by Mi{Trojan-GAL4.0}ChAT[MI04508-TG4.0] CG7715[MI04508-TG4.0-X] and detected in the anterior (cardia/R1) as well as posterior midgut (R4–R5), at the hindgut boundary and rectal ampulla. DNA (blue) is labeled with Hoechst. n=5 guts. N=3. Scale bars 50 μm. (**B**) Stitched confocal immunofluorescence images of a gut expressing GFP

*Figure 3 continued on next page*

Figure 3 continued

(green) under control of Mi{Trojan-GAL4.0}ChAT[MI04508-TG4.0] CG7715[MI04508-TG4.0-X], stained with anti-cut antibody (white). Yellow arrows indicate GFP-positive cells. Enlarged insert shows GFP-positive cells adjacent to the gastric region labeled with cut (pink arrows). DNA (blue) is labeled with Hoechst. n=5 guts. N=3. Scale bar 50 μm. (**B′**) Confocal image of a gut expressing GFP (green) under control of Mi{Trojan-GAL4.0} ChAT[MI04508-TG4.0] CG7715[MI04508-TG4.0-X], stained with Phalloidin (red). Transverse section of the epithelium is shown revealing inter-epithelial axons from ChAT+ neurons. White arrowheads highlight axonal boutons. n=5 guts. N=3. (**C**) Fluorescent immunohistochemistry image of posterior midgut expressing GFP (green) under the control of Mi{Trojan-GAL4.0}ChAT[MI04508-TG4.0] CG7715[MI04508-TG4.0-X], stained with anti-prospero antibody (white). Arrows indicate GFP-positive cells that also label for pros. DNA (blue) is labeled with Hoechst. n=8 guts. N=3. Scale bar 10 μm. (**D**) Confocal immunofluorescence image of ChAT antibody staining of the posterior midgut. EEs are expressing GFP (green) driven by pros-Gal4, yellow arrows indicate the overlap between ChAT staining (white) and pros-positive cells. DNA (blue) is labeled with Hoechst. n=8 guts. N=3. Scale bar 10 μm. (**E**) Barrier dysfunction assay after mCherry (control) or ChAT knockdown in EEs for 3 days with prospero-Gal4. n=100 animals per genotype and condition; N=3. Two-way repeated measures (RM) ANOVA. (**F**) Barrier dysfunction assay after mCherry (control) or ChAT knockdown with Mi{Trojan-GAL4.0}ChAT[MI04508-TG4.0] CG7715[MI04508-TG4.0-X] for 3 days. n=120 animals per genotype and condition; N=3. Two-way RM ANOVA. Data presented as mean ± SEM. ns, not significant, p>0.05; *p≤0.05; **p≤0.01; ***p≤0.001; ****p≤0.0001. n: number of animals or midguts analyzed; N: number of independent experiments performed with similar results and a similar n.

The online version of this article includes the following source data and figure supplement(s) for figure 3:

**Figure supplement 1.** ChAT is expressed in enteric neurons and a subset of EEs.

**Figure supplement 2.** Testing of EE-specific drivers identified from scRNA data and literature.

**Figure supplement 2—source data 1.** Editable table of *Figure 3—figure supplement 2A1*, showing genes highly enriched in EEs.

**Figure supplement 2—source data 2.** Editable table of EE-specific drivers in *Figure 3—figure supplement 2B*, from *Chen et al., 2016*.

**Figure supplement 3.** Testing of EE-specific split-Gal4 drivers without neuronal expression.

In parallel, we analyzed the transcriptome of whole guts depleted of ChAT using pros$^{ts}$ for 3 days. While the knockdown samples also separated clearly from the control, they displayed fewer differentially regulated genes than guts depleted of nAchR (*Figure 4E and F*, *Figure 4—figure supplement 1G*). However, Syt4 remained the most significantly downregulated gene (*Figure 4H*) and enriched GO terms overlapped significantly with the previous experiment, especially with regard to immune responses (*Figure 4G*). Chitin-related GO terms were also trending towards enrichment among downregulated genes (*Figure 4G*), and the genes associated with these terms partially overlapped with the ones identified after nAchR knockdown (*Figure 4C*). We also detected an enrichment of chitin-related gene sets among upregulated genes after ChAT depletion in EEs (*Figure 4—figure supplement 1E*), however the genes associated with these terms were different from the ones previously identified. Direct comparison of differentially regulated genes revealed an overlap of 56 genes between guts depleted for nAchR in ECs and guts where ChAT was silenced using pros$^{ts}$ (*Figure 4I*).

We referenced scRNAseq data recently reported by us in the Aging Fly Cell Atlas (AFCA) (*Lu et al., 2023*) to characterize expression patterns of nAchR signaling components more closely across the different intestinal cell types (*Figure 4—figure supplement 2A, B*). Overall, all nAchR subunits as well as the Ach-producing enzyme ChAT are lowly expressed in the gut epithelium of 5-day-old animals, with nAchR α4 showing the highest expression levels across all cell types. All subunits are expressed in ECs and EEs, while expression in ISCs and EBs is more variable. Subunit α5 shows an enrichment in EEs. ChAT is preferentially expressed in EEs, but also shows some residual EC expression. The Ach-degrading enzyme Ach esterase (Ace) on the other hand is more widely expressed and shows an enrichment in ECs, potentially suggesting a local modulation of Ach levels in the gut epithelium. Finally, Syt4 is expressed at low levels among all intestinal epithelial cell types.

## nAchR depletion disrupts PM integrity

The enrichment of chitin GO terms in our RNAseq experiments prompted us to examine the PM. The PM is a protective structure lining the gut of many insects, consisting of crosslinked glycoproteins, proteoglycans, and chitin (*Erlandson et al., 2019*; *Hegedus et al., 2009*; *Hegedus et al., 2019*). It surrounds the food bolus and forms a selectively permeable physical barrier preventing direct contact between abrasive food particles and bacteria with the epithelium, thus helping to compartmentalize digestive processes as well as protecting the animal from ingested toxins and pathogens (*Erlandson et al., 2019*; *Hegedus et al., 2019*). In flies, it was shown that the PM protects against pathogenic bacteria and their pore-forming toxins, such as *Pseudomonas entomophilia* and *Serratia marcescens* (*Kuraishi et al., 2011*). Dipteran insects such as *Drosophila* are thought to continuously produce a

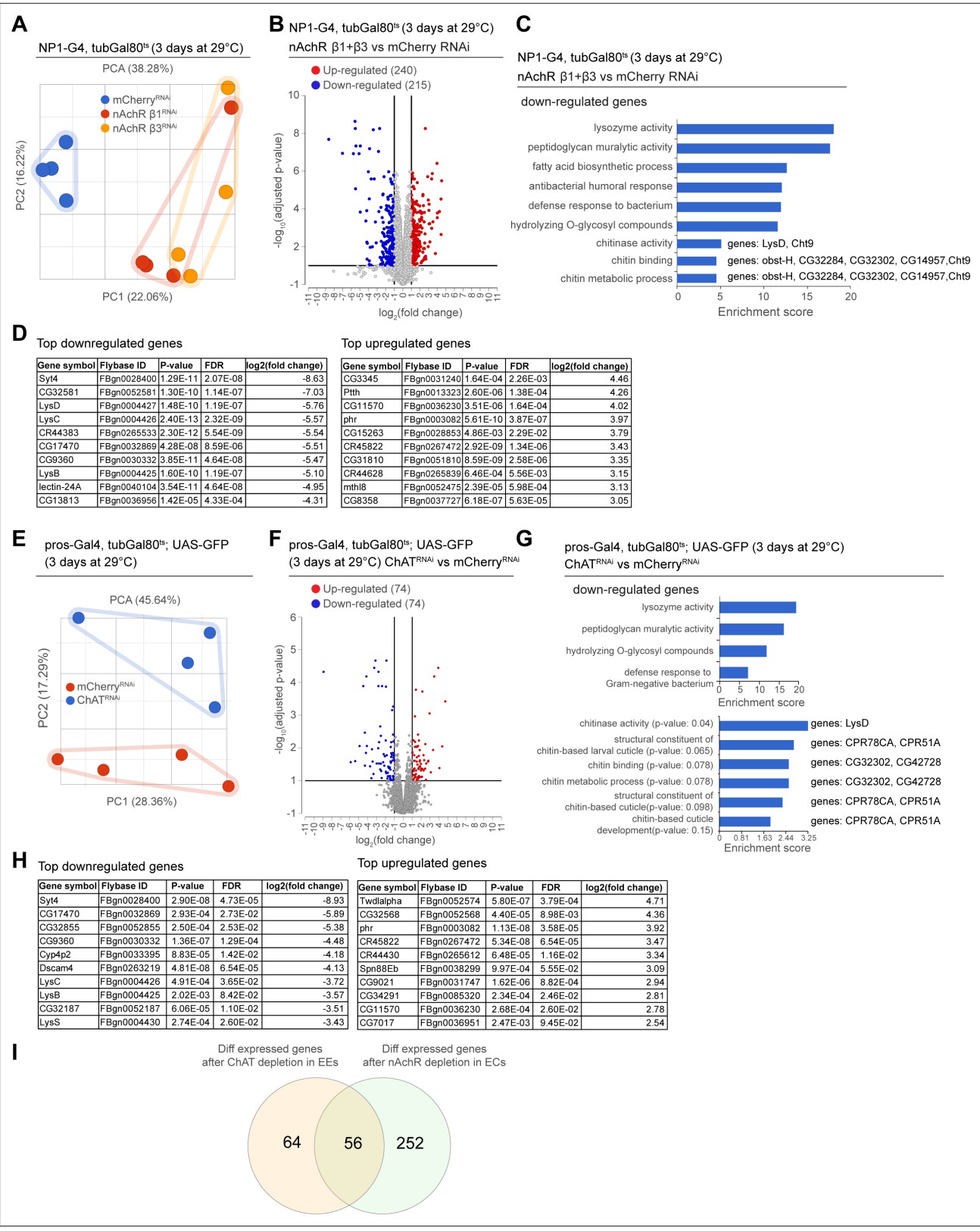

**Figure 4.** Transcriptional changes after disruption of acetylcholine (Ach) signaling in the intestinal epithelium. (**A**) PCA plot of samples after 3 days of nicotinic acetylcholine receptor (nAchR) subunit depletion by RNAi in enterocytes (ECs) with NP1ts. n=4 samples. N=1. (**B**) Volcano plot showing significantly differentially regulated genes after short-term nAchR β1 or β3 knockdown in ECs (FDR ≤ 0.1; log2(fold change) <–1 or >1; 100% of samples have ≥1 reads). (**C**) Gene set enrichment analysis of significantly downregulated genes after nAchR β1 and β3 knockdown in ECs. Genes included in

*Figure 4 continued*

data set associated with chitinase activity, chitin binding, and chitin metabolic processes are listed to the right. (**D**) Top 10 most down- or upregulated genes after 3 days of nAchR subunit depletion by RNAi in ECs with NP1[ts]. (**E**) PCA after 3 days of choline acetyltransferase (ChAT) depletion with RNAi in enteroendocrine cells (EEs) under control of pros[ts]. n=4 samples. N=1. (**F**) Volcano plot of significantly differently regulated genes after 3 days of ChAT knockdown in EEs (FDR ≤ 0.1; log2(fold change) <–1 or >1; 100% of samples have ≥1 reads). (**G**) Gene set enrichment analysis of significantly downregulated genes after ChAT depletion in EEs. Chitin-related terms show a trend toward enrichment. Genes associated with chitin-related terms are listed to the right. (**H**) Top 10 most down- or upregulated genes after 3 days of ChAT knockdown by RNAi in EEs with pros[ts]. (**I**) Overlap between differentially regulated genes after 3 days knockdown of ChAT in EEs or nAchR β1 and β3 in ECs. n: number of samples included; N: number of independent experiments performed with similar results and a similar n.

The online version of this article includes the following source data and figure supplement(s) for figure 4:

**Source data 1.** Editable table of up- and down-regulated genes shown in *Figure 4D*.

**Source data 2.** Editable table of up- and down-regulated genes shown in *Figure 4H*.

**Figure supplement 1.** Transcriptional changes after nAchR depletin in ECs or ChAT depletion in EEs.

**Figure supplement 2.** Expression of genes related to Ach-signaling in scRNAseq data.

type II PM originating in the cardia at the anterior end of the midgut (*Hegedus et al., 2019*). There is evidence suggesting remodeling activity along the posterior midgut, as transcripts for PM components were found enriched in the R4 region of the midgut (*Buchon et al., 2013*). Moreover, intestinal IMD signaling as well as a subset of enteric neurons have been implicated in modulating the composition and permeability of the PM, however the underlying molecular mechanisms of PM remodeling remain poorly understood (*Buchon et al., 2009b*; *Kenmoku et al., 2016*).

Transcript levels of a putative component of the PM, CG32302, were noticeably reduced in guts depleted of nAchR subunits (*Figure 5—figure supplement 1A*). Earlier studies highlighted the importance of the PM in protecting the animal against lethal pathogenic bacterial infection with PE (*Kuraishi et al., 2011*). Indeed, depletion of nAchR subunits in ECs significantly reduced survival after PE infection (*Figure 5A*). Overnight (16 hr) PE infection led to a significant upregulation of nAchR β3 subunit transcript levels, while Ace levels were significantly reduced (*Figure 5—figure supplement 1B*), indicating a possible activation of the pathway in response to infection. ChAT transcript levels as well as numbers of ChAT-expressing cells in the gut epithelium remained unaffected by pathogenic infection (*Figure 5—figure supplement 1B, C*).

Defects in the PM can be visualized with confocal light microscopy by feeding animals fluorescently labeled latex beads that are retained in the food bolus and stay separated from the epithelium if the PM sleeve is intact (*Kenmoku et al., 2016*). The surface of the bead-containing ingested food appeared relatively smooth in control animals. In contrast, silencing of nAchR β1 or β3 led to spiny protrusions of the fluorescent matter, indicating a damaged PM (*Figure 5—figure supplement 1D*). We further modified this assay by crossing in the brush border marker A142-GFP (*Buchon et al., 2013*) and visualizing the PM with fluorescently labeled wheat germ agglutinin (WGA), a lectin that recognizes chitin (*Carlini and Grossi-de-Sá, 2002*), in addition to feeding the latex beads. Guts depleted of nAchR α4 displayed fluorescent signal scattered throughout the lumen and making contact with the brush border while the beads stayed confined to the PM sleeve and separated from the epithelium in control guts (*Figure 5B*).

Electron microscopy has been successfully applied to detect subtle defects in PM morphology (*Kuraishi et al., 2011*). We therefore performed an ultrastructural analysis of the R4 compartment of guts depleted of nAchR β1. The PM was visible as a continuous folded ring in the lumen of control flies, consisting of electron-dense membranous material of roughly 100–200 nm thickness. Additionally, a second, much thinner (15–20 nm) membranous ring-shaped layer was observed between the PM and the apical surface of the epithelial cells (*Figure 5C*, *Figure 5—figure supplement 1E*).

In the majority of the 18 examined nAchR β1 knockdown midguts the thick PM layer was not compromised (*Figure 5—figure supplement 1F*). However, in 56% of samples the thin layer was clearly disrupted or missing altogether (*Figure 5C*). Notably, none of the guts presented an intact thin layer in the absence of the thick layer. In all examined control and knockdown guts, the septate junctions connecting adjacent cells appeared normal, consistent with our Dlg and coracle staining (*Figure 2H*, *Figure 2—figure supplement 2A*), as well as the fact that no changes in junctional protein expression was observed in our RNAseq experiments.

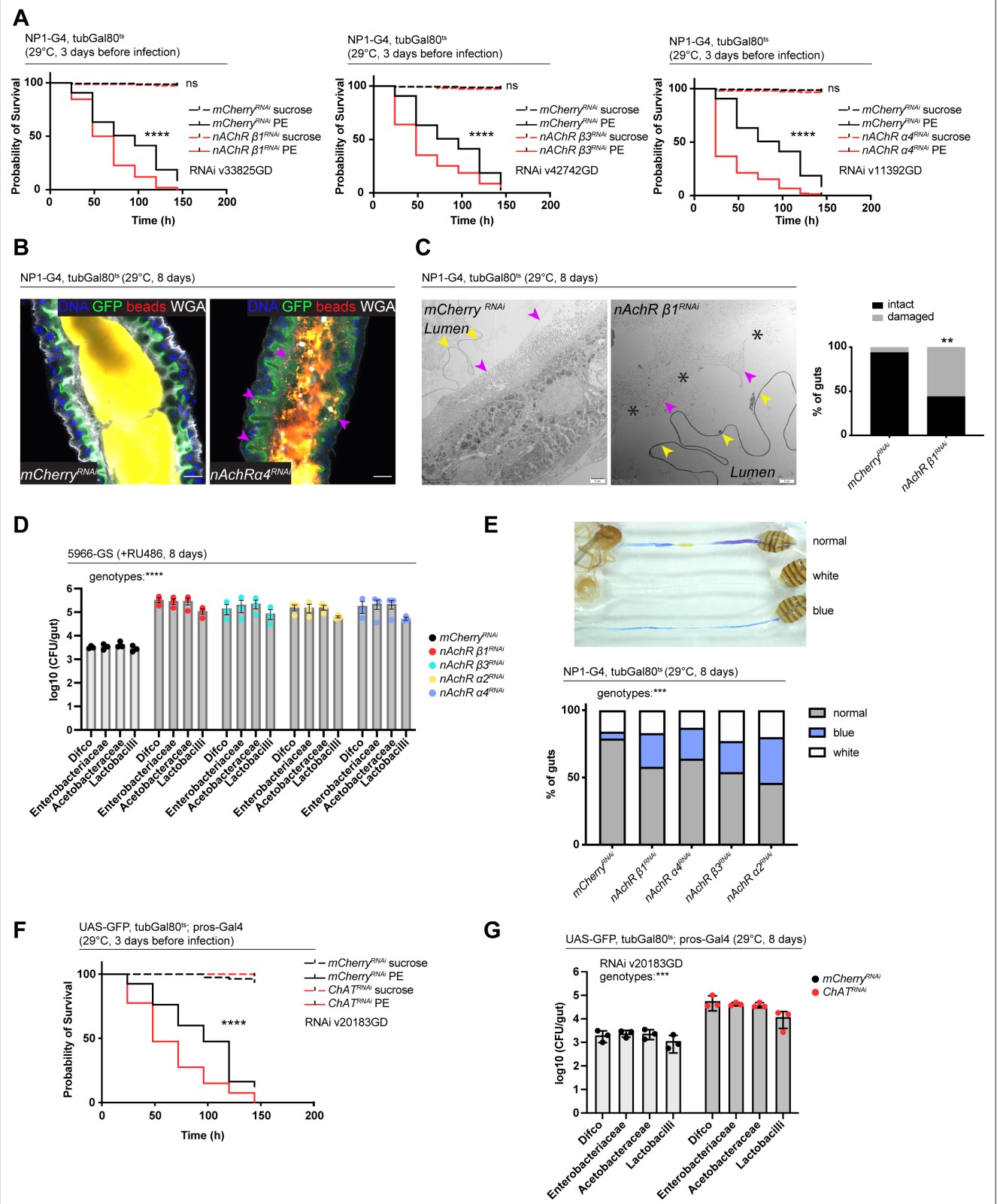

**Figure 5.** Nicotinic acetylcholine receptor (nAChR) depletion disturbs peritrophic matrix (PM) integrity, causes dysbiosis and inflammation. (**A**) Survival of animals depleted for mCherry (control) or nAChR β1, β3, or α4 for 3 days with NP1ᵗˢ before *P. entomophila* infection. n=150 animals per genotype and condition; N=3. Log rank (Mantel-Cox) test. (**B**) Confocal immunofluorescence image of posterior midguts depleted for either mCherry (control) or nAChR α4 for 8 days with NP1ᵗˢ. Animals are expressing a GFP-brush border marker (green) and were fed red fluorescent beads to assess PM

*Figure 5 continued on next page*

*Figure 5 continued*

integrity (beads appear yellow/orange due to autofluorescence of beads in GFP channel). PM is labeled with wheat germ agglutinin (WGA) (white), DNA (blue) is labeled with Hoechst. Pink arrowheads highlight beads no longer contained by the PM sleeve. n=15 guts per genotype. N=3. Scale bar 20 µm. (**C**) Electron microscopy images and quantification of thin PM layer integrity. Thick (yellow arrows) and thin (pink arrows) PM layers are indicated. Asterisks highlight gaps in the thin layer after nAchR β1 depletion. n=16; 18 midguts. N=1. Fisher's exact test. (**D**) Colony forming units (CFUs) of whole guts plated on selective growth media after 8 days of nAchR subunit depletion with 5966-GS. Three pooled independent experiments are shown. n=5 pooled animals per genotype and experiment. Two-way ANOVA. (**E**) Gut compartmentalization and acidity after mCherry (control) or nAchR β1, β3, or α4 depletion for 8 days with NP1ᵗˢ. Healthy flies fed with Bromphenol blue pH indicator display an acidic patch (yellow), while loss of gut compartmentalization leads to all blue or white guts. n=87 guts for mCherry RNAi (control), n=93 guts for nAchR β1 RNAi, n=113 guts for nAchR α4 RNAi, n=87 guts for nAchR β3 RNAi and n=90 guts for nAchR α2 RNAi. Four independent pooled experiments are shown. Chi square test. (**F**) Survival after 3 days of mCherry (ctrl) or choline acetyltransferase (ChAT) depletion in enteroendocrine cells (EEs) followed by *P. entomophila* infection. n=80 animals per genotype and condition; N=3. Log rank (Mantel-Cox) test. (**G**) CFUs of whole guts plated on selective growth media after 8 days of ChAT depletion in EEs. Three pooled independent experiments are shown. n=5 pooled animals per genotype and experiment. Two-way ANOVA. Data presented as mean ± SEM. ns, not significant, p>0.05; *p≤0.05; **p≤0.01; ***p≤0.001; ****p≤0.0001. n: number of animals or midguts analyzed; N: number of independent experiments performed with similar results and a similar n.

The online version of this article includes the following figure supplement(s) for figure 5:

**Figure supplement 1.** Ach signaling modulates PM integrity and infection sensitivity.

## Defects in Ach signaling disturb gut compartmentalization and cause dysbiosis and JAK-STAT-mediated inflammation

Since the PM has been connected to regulation of the microbiome in mosquitoes (*Rodgers et al., 2017*), we hypothesized that nAchR silencing also deregulates the microbial community inhabiting the fly gut. To test this assumption, we measured microbial load by plating pooled guts of control or nAchR knockdown animals on selective media supporting the growth of commensals such as Lactobacilli, Acetobacteriaceae, or Enterobacteriaceae. The amount of CFUs after 8 days of nAchR β1, β3, or α4 silencing exceeded control levels significantly, indicating that these flies struggle to maintain appropriate commensal numbers (*Figure 5D*). The fly midgut is functionally compartmentalized and contains a stomach-like region of acid-producing copper cells (*Dubreuil, 2004*). A previous study has highlighted the importance of gut compartmentalization in controlling microbiome abundance and distribution. As flies age, this spatial organization is progressively lost due to chronic JAK-STAT activation leading to metaplasia of copper cells in the acidic gastric region, ultimately resulting in dysbiosis and death of the animal (*Li et al., 2016*). Gut compartmentalization can be visualized by feeding flies the pH indicator Bromophenol blue, which labels the acidic copper cell region in yellow, while the rest of gut remains blue, indicative of a more basic pH (*Li et al., 2016*). Reduction of nAchR levels lead to an increase of disturbed acidity patterns, ranging from completely blue guts to samples with weak staining and white patches, which has been attributed to expansion of acid-producing commensals like *Lactobacillus* along the whole gut (*Li et al., 2016*; *Figure 5E*).

Dysbiosis can be a consequence of IMD pathway disruption, but at the same time triggers chronic IMD pathway activation (*Buchon et al., 2009a*; *Guo et al., 2014*). Surprisingly, we did not observe an upregulation of antimicrobial peptides transcripts classically associated with an IMD response (*De Gregorio et al., 2002*; *Imler and Bulet, 2005*), which is in line with the enrichment of immune-related terms among downregulated genes in our bulk RNAseq data sets (*Figure 4C and G*).

Furthermore, depletion of ChAT in EEs with prosᵗˢ also caused increased susceptibility to PE infection (*Figure 5F*) as well as dysbiosis (*Figure 5G*). Conversely, overexpression of ChAT promoted survival after bacterial infection (*Figure 5—figure supplement 1G*).

Together these results suggest that Ach signaling is required to maintain a healthy microbiome and protect the animals against pathogenic infections.

## Syt4 is a transcriptional target of nAchR regulating PM function

One of the most significantly downregulated genes identified in our RNAseq data sets is Synaptotagmin 4, a vesicular Ca²⁺-binding protein promoting retrograde signaling at synapses (*Yoshihara et al., 2005*). RNAi-mediated silencing of Syt4 under control of NP1ᵗˢ reduced survival after challenge with PE (*Figure 6A*, *Figure 6—figure supplement 1A*) and caused PM defects visualized with the bead feeding assay (*Figure 6B*). Moreover, Syt4 depletion increased commensal numbers (*Figure 6—figure supplement 1B*), disrupted gut compartmentalization (*Figure 6—figure supplement 1C*), as

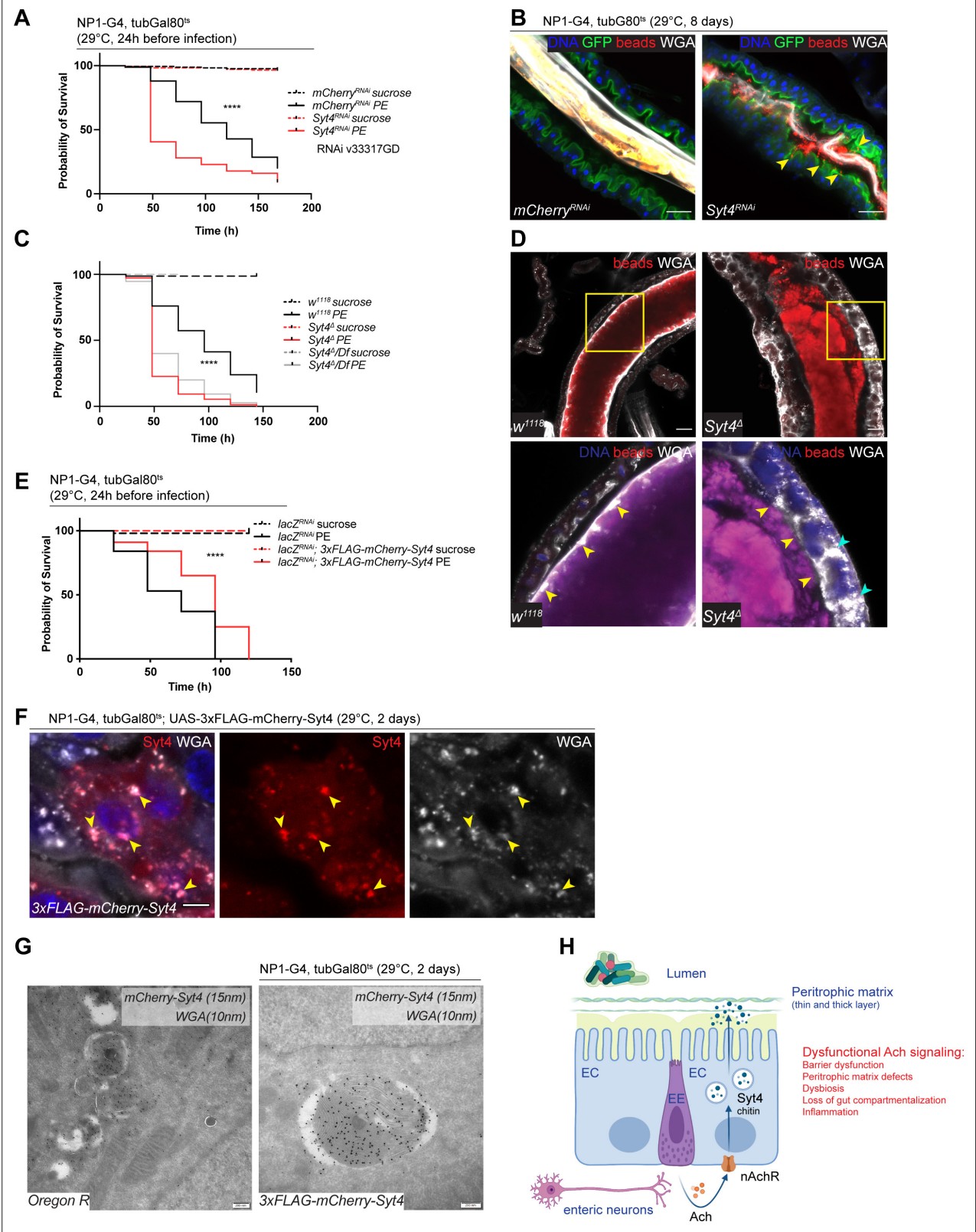

**Figure 6.** Syt4 knockdown affects peritrophic matrix (PM) integrity and phenocopies nAchR depletion. (**A**) Survival after 1 day of mCherry (ctrl) or Syt4 depletion in ECs followed by *P. entomophila* infection. n=175 animals per genotype and condition; N=3. Log rank (Mantel-Cox) test. (**B**) Confocal immunofluorescence image of posterior midguts depleted for either mCherry (control) or Syt4 for 8 days. Animals are expressing a GFP-brush border marker (green) and were fed red fluorescent beads to assess PM integrity (beads appear yellow/orange due to autofluorescence of beads in GFP

*Figure 6 continued on next page*

*Figure 6 continued*

channel). PM is labeled with wheat germ agglutinin (WGA) (white), DNA (blue) is labeled with Hoechst. Yellow arrowheads indicate beads that leaked out of the PM sleeve. n=10 guts per genotype. N=3. Scale bar 25 μm. (**C**) Survival of w1118 (control), Syt4$^\Delta$ CRISPR null mutant flies or Syt4$^\Delta$ flies crossed to a deficiency (BL24927) after *P. entomophila* infection. n=75 animals per genotype and condition; N=3. Log rank (Mantel-Cox) test. (**D**) Confocal immunofluorescence image of posterior midguts of w1118 (control) or Syt4$^\Delta$ animals fed with red fluorescent beads to monitor PM integrity. PM is stained with WGA (white), DNA (blue) is labeled with Hoechst in bottom panels. Yellow insets are shown enlarged in bottom row. Yellow arrowheads indicate the presence (w1118) or absence of a clear PM boundary. Cyan arrowheads indicate accumulation of WGA signal within the epithelium. n=10 guts per genotype. N=3. Scale bar 25 μm. (**E**) Survival after overexpression of LacZ-RNAi (control) or LacZ-RNAi together with UAS-FLAG-mCherry-Syt4 for 1 day before *P. entomophila* infection. n=100 animals per genotype and condition; N=3. Log rank (Mantel-Cox) test. (**F**) Confocal immunofluorescence image of posterior midguts overexpressing UAS-FLAG-mCherry-Syt4 (red) in enterocytes stained with WGA (white). DNA (blue) is labeled with Hoechst in bottom panels. Yellow arrowheads indicate overlap between Syt4-positive vesicles and WGA staining. n=8 guts. N=3. Scale bar 25 μm. (**G**) Immunogold electron microscopy image of posterior midgut of an Oregon R wildtype animal or an animal overexpressing UAS-FLAG-mCherry-Syt4 in enterocytes with NP1ts. WGA-biotin (10 nm gold particles) is detected in multilamellar bodies carrying membranous and amorphous material. Syt4 (stained with anti-mCherry antibody, 15 nm gold particles) colocalizes with these structures in animals expressing the UAS-FLAG-mCherry-Syt4, while Oregon R samples are devoid of anti-mCherry antibody labeling. n=5. N=1. Scale bar 200 nm. (**H**) Model: Neuronal or EE-derived Ach maintains barrier function through Syt4-mediated secretion of PM components such as chitin from ECs. Disrupted Ach signaling leads to barrier dysfunction, PM defects, dysbiosis, as well as loss of gut compartmentalization and inflammation. Ach, acetylcholine; nAchR, nicotinic acetylcholine receptor; EC, enterocyte; EE, enteroendocrine cell; Syt4, Synaptotagmin 4. Data presented as mean ± SEM. ns, not significant, p>0.05; *p≤0.05; **p≤0.01; ***p≤0.001; ****p≤0.0001. n: number of animals or midguts analyzed; N: number of independent experiments performed with similar results and a similar n.

The online version of this article includes the following figure supplement(s) for figure 6:

**Figure supplement 1.** Syt4 promotes survival after infection, a healthy microbiome and gut compartmentalization but is not sufficient to rescue loss of nAchR in ECs.

well as the morphology of the gastric region: Acid-producing copper cells usually form deep invaginations of the apical membrane (*Dubreuil, 2004*), giving rise to a gastric unit that can be visualized with anti-cut staining (*Li et al., 2016*). Syt4 depletion resulted in a disorganized morphology and a marked flattening of these units (*Figure 6—figure supplement 1D*). Combined knockdown of Syt4 and nAchR subunits did not have an additive effect on survival after PE infection, consistent with an epistatic relationship between these genes (*Figure 6—figure supplement 1E*).

We generated a new Syt4 null mutant (Syt4$^\Delta$) with CRISPR/Cas9 technology to further substantiate these findings. While these mutant animals were homozygous viable, they displayed enhanced susceptibility to PE challenge (*Figure 6C*) and a fragmented PM, often accompanied by enlarged WGA-positive structures within the epithelium (*Figure 6D*). To control for potential off-target effects of the CRISPR/Cas9 technology, we outcrossed Syt4$^\Delta$ to a deficiency covering the Syt4 locus and found that the resulting offspring were also highly sensitive to PE infection (*Figure 6C*) and displayed barrier dysfunction after bacterial challenge (*Figure 6—figure supplement 1F*). The response of these mutants to Paraquat stress was less robust: Syt4 mutant animals were slightly (but not significantly) more sensitive to Paraquat than wildtype controls (*Figure 6—figure supplement 1G*). Accordingly, RNAi-mediated knockdown of Syt4 in ECs resulted in overall mild and experimentally inconsistent increases in barrier dysfunction after Paraquat treatment (*Figure 6—figure supplement 1H*).

To characterize the localization of Syt4 in the gut epithelium, we utilized a 3xFLAG-mCherry-labeled protein trap line under UAS control (*Singari et al., 2014*). Overexpression of this construct with NP1$^{ts}$ had a protective effect on animal survival after PE challenge (*Figure 6E*), but was not able to rescue PE sensitivity when nAchR subunits were knocked down simultaneously, suggesting that Syt4 expression is not sufficient for barrier protection in an nAchR loss of function context (*Figure 6—figure supplement 1I*). Immunofluorescence analysis of the overexpressed construct yielded a vesicular staining pattern that overlapped with Golgi and lysosomal markers (*Figure 6—figure supplement 1J*). Interestingly, we noticed a significant colocalization of Syt4-mCherry-positive structures and WGA staining in immunofluorescence experiments, suggesting that Syt4 vesicles contain chitin (*Figure 6F*). Immunogold electron microscopy confirmed the colocalization of mCherry-Syt4 and WGA in vesicular structures containing highly folded membrane swirls and amorphous cargo (*Figure 6G*). These vesicles also stained positive for the late endosomal/lysosomal marker Lamp1 (*Figure 6—figure supplement 1K*). Similar WGA or Lamp1 carrying structures were observed in wildtype (OreR) guts (*Figure 6G* and *Figure 6—figure supplement 1K*).

## Discussion

Our study identified 22 candidate COPD genes that modulate barrier integrity in the fly and demonstrate the utility of the genetically accessible fly to screen candidate disease genes from human GWAS to provide mechanistic insight into their role in tissue homeostasis and pathophysiology. In particular, our results provide a role for nAchR signaling in maintaining intestinal barrier function. Since depletion of several nAchR subunits in ECs, or of ChAT in neurons and EEs leads to loss of barrier integrity and decreased survival after chemical or bacterial challenge, we propose that Ach-mediated crosstalk between cholinergic neurons and/or EEs with ECs is critical to maintain intestinal epithelial homeostasis. This role of nAchR signaling is mediated by transcriptional regulation of Syt4 in ECs, which in turn maintains secretion of chitin vesicles to maintain the PM (*Figure 6H*).

The exact source of Ach in this regulation remains unclear due to the lack of specific drivers that would allow separating cholinergic neurons and EEs. Recent work suggests a vital role of neuronally derived Ach during regeneration (*Petsakou et al., 2023*). While neuronal Ach seems to be the main driver regulating return to homeostasis after injury, it is possible that Ach secreted by both cell types may contribute to the maintenance of the epithelium under homeostatic conditions and it will be of interest to explore these contributions in future studies.

Of further interest for future studies is to assess the regulation of nAchR signaling activity in the context of stress and infection. Our transcriptome analysis, which shows that nAchR β3 is induced and Ace is repressed after PE infection, is in line with previous findings and suggests that the pathway is dynamically regulated in response to enteropathogen infection. In addition to regulating tissue recovery, $Ca^{2+}$ signaling in ECs may also strengthen PM integrity through promoting Syt4-mediated vesicle fusion (*Wang and Chapman, 2010*).

Our data using nAChR subunit loss of function clones suggest that nAchR signaling only mildly impacts regenerative processes, influencing the balance between EEs and ECs in newly formed cell populations. It is possible that perturbing other subunits may have a stronger impact on ISC proliferation and/or differentiation and an exhaustive loss of function study evaluating the role of specific subunits will be of interest. We do observe a strong proliferative response of ISCs to nAchR subunit knockdown in ECs, on the other hand, and this response is consistent with previously described proliferative responses of ISCs to EC stress and damage (*Jiang et al., 2009*).

A role for Ach signaling in human barrier epithelia is supported by previous studies: muscarinic Ach receptors have been successfully targeted in the clinic to relieve bronchoconstriction and mucus hypersecretion in COPD and asthma (*Calzetta et al., 2021*), although nicotinic AchRs remain more elusive from a therapeutic perspective (*Hollenhorst and Krasteva-Christ, 2021*). While Ach is a classic neurotransmitter, a growing body of work has uncovered an important role of Ach beyond the context of the nervous system: various non-neuronal cell types express the machinery for Ach synthesis and secretion, ranging from diverse immune cells to epithelial cells, such as brush/tuft cells (*Kummer and Krasteva-Christ, 2014*; *Wessler and Kirkpatrick, 2008*). Airway tuft cells have been implicated in orchestrating type 2 inflammatory responses (*Parker et al., 2019*; *Sell et al., 2021*) and mucociliary clearance (*Perniss et al., 2020*), whereas their intestinal counterparts participate in defense against helminths and protists and limit biliary inflammation (*O'Leary et al., 2019*; *Parker et al., 2019*). With a wide range of cell types able to produce or sense Ach, non-neuronal Ach serves as a versatile signaling molecule eliciting complex intercellular crosstalk with diverse outcomes; depending on the context, Ach may promote inflammation or conversely exert anti-inflammatory functions (*Hollenhorst and Krasteva-Christ, 2021*; *Kummer and Krasteva-Christ, 2014*; *Sell et al., 2021*). Accordingly, it was recently reported that expression of a COPD risk allele of CHRNA5 in epithelial cells leads to airway remodeling in vivo, increased proliferation and production of pro-inflammatory cytokines through decreased calcium entry and increased adenylyl-cyclase activity (*Routhier et al., 2021*).

Previous work has further shown a downregulation of junctional proteins such as ZO-1 and p120 after depletion of CHRNA5 in A549 lung cancer cells (*Krais et al., 2011*). While we observed a mildly disorganized pattern of junctional markers such as Dlg after nAchR subunit knockdown in the fly intestinal epithelium, junctional architecture appeared normal when analyzed by electron microscopy. Furthermore, transcriptome analysis revealed little to no changes in the expression of proteins involved in polarity or cellular junction formation, suggesting that nAchR signaling regulates barrier function through other mechanisms. A role for the Syt4-mediated secretion of PM protein components

and chitin in maintaining barrier integrity is supported by the observation that mCherry-tagged Syt4 partially overlaps with chitin-binding WGA staining.

While the *Drosophila* PM is thought to be produced mostly in the anterior most portion of the gut (*Hegedus et al., 2019*), the existence of WGA-positive vesicles throughout the entire midgut suggests continuous remodeling along the length of the tissue. This finding is consistent with the previously reported expression of PM-related transcripts in the R4 compartment of the midgut (*Buchon et al., 2013*). PM integrity can be modulated by enteric neurons, although a role for cholinergic signaling was not tested in this context (*Kenmoku et al., 2016*).

Our study highlights the evolutionary conservation of mechanisms maintaining epithelial barrier function. The PM is functionally analogous to mucus and surfactant layers in mammalian airways, and it remains to be explored whether COPD risk alleles in nAchR subunits also cause a dysfunction in the secretion of such barrier components. The elevated inflammation and airway remodeling in mice expressing the CHRNA5 risk allele suggest that such a mechanism may be conserved as well (*Routhier et al., 2021*). It is critical to note that the epithelial dysfunction observed in these animals, as well as part of the association of COPD risk with specific CHRNA loci, emerges independently of cigarette smoke (*Parker et al., 2019*; *Routhier et al., 2021*; *Siedlinski et al., 2013*), indicating that nAchR signaling is critical to maintain homeostasis not only in the context of oxidative stress, but under homeostatic conditions. Supporting this view, our data show that knockdown of nAchR subunits in fly ECs also results in epithelial stress signaling in the absence of Paraquat exposure.

These consistencies further validate the approach of prioritizing candidate genes associated with COPD risk loci using the *Drosophila* intestine as a model system. Characterization of the epithelial role of other identified candidate orthologs from our screen will likely provide further insight into the biology and pathophysiology of barrier dysfunction and epithelial homeostasis. Such studies will be critical for target identification and validation for therapeutic intervention in COPD.

# Materials and methods
## *Drosophila* stocks and husbandry

Flies were raised and kept on standard fly food prepared according to the following recipe: 1 l distilled water, 13.8 g agar, 22 g molasses, 75 g malt extract, 18 g inactivated dry yeast, 80 g corn flour, 10 g soy flour, 6.26 ml propionic acid, 2 g methyl 4-hydroxybenzoate in 7.2 ml of ethanol. Flies were reared at 25°C with 65% humidity on a 12 hr light/dark cycle. All animals used in this study were mated females matured for 4–6 days.

The TARGET system was used to conditionally express UAS-linked transgenes in specific cell populations in combination with indicated Gal drivers (*McGuire et al., 2004*).

Crosses containing tub::Gal80$^{ts}$ were reared at 18°C to avoid premature gene expression. Transgene expression was induced by shifting the flies to 29°C for 1–8 days, as indicated in the figure legends.

For experiments using a GeneSwitch driver, flies were reared on normal food before being shifted to food containing 200 mM Mifepristone (RU486); for barrier function experiments (smurf assay). FD&C blue dye (Neta Scientific, SPCM-FD110-04) was added to the food at final concentration of 2.5% (wt/vol).

Formation of MARCM clones was induced with heat shock for 1 hr at 37°C and clones were analyzed after 7 days at 25°C. EsgF/O clones were analyzed after 8 days at 29°C.

RNAi lines used in the barrier dysfunction screen are listed in *Supplementary file 2*.

The following additional lines were obtained from the Bloomington Drosophila Stock Center: w$^{1118}$ (3605), Oregon-R (5), UAS-mCherry$^{RNAi}$ (35785), UAS-ChAT$^{RNAi}$ (60028), FRT82B (2051), TI{TI}nAChRα2$^{attP}$ (84540), Mi{Trojan-GAL4.0}ChAT[MI04508-TG4.0] CG7715[MI04508-TG4.0-X]/TM6B (60317), TI{2A-GAL4}ChAT[2A-GAL4] (84618), Orcokinin-Gal4 (92253), arm-GFP (8555), Df(3R)BSC423/TM6C (24927), CG32547-Gal4/FM7a (84614), UAS-Luciferase$^{RNAi}$(31603), nAChRalpha1$^{attP}$/TM6B (84539), UAS-Syt4$^{RNAi}$ (39016), Mi{Hto-WP}GldGYB Syt4GYB/TM6B (56539), UAS-2xEGFP (6874), dimm-Gal4 (25373), CCAP-Gal4 (25686), CCAP-Gal4 (25685), Burs-Gal4 (51980), Dsk[2A]-Gal4 (84630), Dsk-Gal4 (51981), Nplp4[2A]-Gal4 (84674), R57F07-p65.AD; UAS-DSCP-6XEGFP (91402), UAS-DSCP-6XEGFP; R57F07-Gal4.DBD (91403), R33A12-Gal4.DBD (68537), R61H08-Gal4.DBD (69158), Mi{Trojan-Gal4.DBD.0}ChAT[MI04508-TG4DBD.0] CG7715[MI04508-TG4DBD.0-X] (60318).

The following additional lines were obtained from the Vienna *Drosophila* Stock Center: UAS-nAchR β1[RNAi] (33825, pruned), UAS-nAchR β3[RNAi] (42742, pruned), UAS-ChAT[RNAi] (20183), UAS-Syt4[RNAi] (v33317).

The following lines were gifts: 5966-GeneSwitch (B Ohlstein), 2xSTAT::GFP (E Bach), NP1-Gal4 (D Ferrandon), A142-GFP (N Buchon), Mex1-Gal4;tubGal80[ts] (L O'Brien), MARCM82 (hsFlp; tubGal4, UAS-GFP; FRT82, tubGal80, N Perrimon), ProsV1-Gal4 (J-F Ferveur), da-GS (V Monnier), UAS-Xbp1-eGFP (H D Ryoo), esg-Gal4, UAS-GFP, tubGal80[ts]; UAS-Flp, Actin>CD2>Gal4 (esgF/O, B Edgar).

## Generation of UAS-ChAT

DNA encoding the sequence of choline *O*-acetyltransferase (Uniprot identifier P07668, amino acid residues 1–721) was synthesized and subcloned into pUASTattB under the control of the hsp70 promoter. Transgenic lines were established by WellGenetics, Taiwan. In brief, pUASTattB plasmid containing the ChAT sequence was microinjected into embryos of y[1] M{vas-int.Dm}ZH-2A w[*]; P{y[+t7.7]=CaryP} attP40 or y[1] M{vas-int.Dm}ZH-2A w[*]; P{y[+t7.7]=CaryP}attP2. Transgenic F1 flies were screened for the selection marker white+ (orange colored eyes).

## Syt4 CRISPR mutant

CRISPR-mediated mutagenesis was performed by WellGenetics, Inc (Taiwan) using modified methods of *Kondo and Ueda, 2013*. In brief, the upstream gRNA sequences TTTCCACTCGATGTTCCTGG [CGG] and downstream gRNA sequences CGCAGGCGCCCCTTAATGAG[GGG] were cloned into U6 promoter plasmids separately. Cassette 3xP3 RFP, which contains a floxed 3xP3 RFP and two homology arms, were cloned into pUC57 Kan as donor template for repair. Syt4/CG10047-targeting gRNAs and hs Cas9 were supplied in DNA plasmids, together with donor plasmid for microinjection into embryos of control strain w[1118]. F1 flies carrying the selection marker 3xP3 RFP were further validated by genomic PCR and sequencing. This CRISPR editing generates a 2603 bp deletion allele of Syt4, deleting the entire CDS and replacing it with a 3xP3 RFP cassette.

## Gene assignment to COPD GWAS loci

Publicly available summary statistics for the discovery stage of the COPD GWAS reported by *Hobbs et al., 2017*, were obtained from dbGaP (accession: phs000179). Forty-eight candidate genes were assigned to the 22 loci reported in *Hobbs et al., 2017*, based upon eQTL, coding variation level support, or physical distance if a gene could not be assigned via the former criteria. First, candidate genes were assigned to loci if the index variant was an eQTL in any tissue for any gene within 250 kilobases of the variant in GTEx (*Battle et al., 2017*) (V6p). We further applied colocalization (via the coloc package in R) (*Giambartolomei et al., 2014*) to estimate the probability the eQTL and COPD risk association signal share a casual variant. Of the 40 genes with eQTL support, 24 had a colocalization probability >0.6. Candidate genes were also assigned to loci if the index variant was in linkage disequilibrium (LD) ($r^2 > 0.6$) with coding variants for the gene. LD was estimated using individuals of European ancestry from 1000 Genomes (*Auton et al., 2015*). Eight candidate genes were assigned to five loci, six of which overlapped genes with eQTL level support.

Since we first obtained this candidate gene list, a larger COPD risk GWAS was published (*Sakornsakolpat et al., 2019*) that made use of not only lung eQTL and coding variant data, but also epigenetic and gene set similarity approaches to assign candidate genes to COPD risk loci (see Supplementary Table 7 in *Sakornsakolpat et al., 2019*). We found our assigned candidate genes overlapped with candidate genes from this newer study at 13/22 loci reported in the *Hobbs et al., 2017*, study, including CHRNA3/5. Overall 20/48 candidate genes were also listed as candidate genes in the *Sakornsakolpat et al., 2019*, study.

## Barrier dysfunction screen

For the barrier dysfunction assay, males from candidate RNAi lines were crossed at a 1:1 ratio with virgin *daughterless*-GeneSwitch driver line females in Bloomington-modified food (standard medium) bottles. Crosses were raised at 25°C and brooded every 2–3 days. Progeny were collected and females were sorted after mating for 2–3 days (discarding males). This yielded about 200 females per genotype depending on the RNAi line. Sorted females were aged in standard medium at 25°C for 10–12 days. Aged females (~25 per vial; exact number recorded per vial for assay readout) were

then exposed to standard medium prepared with 200 mM Mifepristone (RU486) from Sigma-Aldrich (cat# 856177) and 2.5% wt/vol FD&C Blue Dye no 1 from Spectrum (cat# FD110) for 24 hr at 25°C or 29°C. Prior to Paraquat exposure, flies were dry starved for 2–3 hr at the experimental temperature. 25 mM Paraquat solution (5% sucrose and 2.5% wt/vol FD&C Blue Dye in sterile water) or mock solution (sucrose and blue dye only) was freshly prepared for each experiment. Starved flies were placed in empty vials with a Whatman filter paper (VWR, 89013-946) on top of a foam biopsy pad (Neta Sciences, BPLS-6110) and 1.25 ml of Paraquat or mock solution for 16 hr at 25°C or 29°C and then shifted back to medium with 200 mM Mifepristone (RU486) and 2.5% wt/vol FD&C Blue Dye no 1. Entirely Blue (Smurf) flies were counted starting post-16-hr exposure. Smurf flies were counted daily or every other day. About 8–12 candidate RNAi lines were tested in sets with Luciferase RNAi always included as a control.

The average proportion of smurf flies across technical replicates per time point were calculated and graphed. The natural log (LN) ratio was calculated for each candidate RNAi by dividing the candidate RNAi proportion average from the final time point by the Luciferase RNAi proportion average for the same time point (=LN(Candidate RNAi/Luciferase RNAi)). Candidate RNAi results were ranked by establishing a scale with arbitrary LN ratio ranges to define: strong enhancers ($\geq$1), enhancers ($\geq$0.43 to $\leq$0.99), weak enhancers ($\geq$0.20 to $\leq$0.42), no effect ($\geq$–0.30 to $\leq$0.19), weak suppressors ($\geq$–0.70 to $\leq$–0.31), suppressors ($\geq$–1.10 to $\leq$–0.71), and strong suppressors ($\leq$–1.11).

## Paraquat feeding

20–25 flies per vial were kept on food containing FD&C blue for 1–3 days and dry starved in empty vials for 2–3 hr prior to Paraquat exposure. Methyl viologen dichloride hydrate (Paraquat, 856177, Sigma-Aldrich) solution was prepared freshly for each experiment in 5% (wt/vol) sucrose in water with 2% (wt/vol) FD&C blue. Paraquat concentration was 12.5 mM unless indicated otherwise. Starved flies were transferred to vials containing 600 µl of Paraquat solution or 5% sucrose (mock treatment) as well as a circular Whatman filter paper (VWR, 89013-946) on top of a foam biopsy pad (Neta Sciences, BPLS-6110). Flies were treated for 16 hr overnight and then transferred back to fly food with FD&C blue dye. The number of smurf flies was recorded 24 hr after the start of the Paraquat challenge and subsequently monitored over the course of 7–10 days.

## PE infection

PE (gift from B Lemaitre) was cultured in LB medium at 29°C overnight for 16–18 hr (15 ml/sample to be infected). Bacteria were centrifuged at 4000×$g$ for 10 min at room temperature (RT) and resuspended in 5% sucrose (OD$_{600}$=100). 500 µl of concentrated bacterial suspension or 5% (wt/vol) sucrose solution (mock treatment) was added to empty fly vials containing a circular Whatman filter paper (VWR, 89013-946) on top of a foam biopsy pad (Neta Sciences, BPLS-6110). For assessment of barrier function, bacteria were suspended in 5% sucrose+2% wt/vol FD&C Blue Dye no 1. 20–25 flies per vial were starved in empty vials for 2–3 hr before infection. Survival was monitored over the course of 7–10 days and 100 µl of 5% sucrose or 5% sucrose+2% wt/vol FD&C Blue Dye no 1 solution was added daily.

## Gut compartmentalization

Gut compartmentalization was assessed as described in *Li et al., 2016*: 100 µl of 2% wt/vol Bromphenol blue solution (Sigma-Aldrich, B5525) was dispensed in a food vial, the surface was broken up with a pipette tip to allow full absorption of the dye before flies were transferred onto food. Flies were fed overnight and guts were dissected in small groups and immediately scored visually under a stereomicroscope to avoid prolonged exposure to $CO_2$.

## Immunofluorescence microscopy

Guts from adult female flies were dissected in PBS, fixed for 45 min at RT in fixative solution (4% formaldehyde, 100 mM glutamic acid, 25 mM KCl, 20 mM MgSO$_4$, 4 mM Na$_2$HPO$_4$, 1 mM MgCl$_2$, pH 7.5), washed twice in wash buffer (1× PBS, 0.5% bovine serum albumin and 0.1% Triton X-100, 0.005% NaN$_3$) for 30 min at RT. Primary and secondary antibodies were diluted in wash buffer. Samples were incubated overnight at 4°C with primary antibody, washed twice for 30 min with wash buffer before incubating 4–6 hr at RT with secondary antibody. Hoechst33342 (Invitrogen, H3570, 1:10,000) or

WGA-AlexaFluor647 (Invitrogen, W32466, 1:500) were added to the secondary antibody cocktail to visualize DNA or the PM respectively. Samples were washed again twice for 30 min before mounting in Prolong Glass antifade mounting media (Invitrogen, P36982).

To assess the integrity of the PM, flies were dry starved for 2 hr and then fed Fluoresbrite microspheres (Polysciences, 17149 [0.05 μm, green] or 195075 [0.5 μm, red]), diluted 1:50 in 5% sucrose solution on Whatman filter paper for 16 hr overnight. Guts were dissected, fixed, and processed for immunofluorescence microscopy analysis as described above.

For lysotracker staining, freshly dissected guts were incubated for 5 min in 1× PBS with Lysotracker Deep Red (Invitrogen, L12492, 1:500) before fixation. Samples were washed twice for 10 min before and after 1 hr incubation with Hoechst, mounted and analyzed within 1 day.

Primary antibodies used in this study: chicken anti-GFP (Abcam, ab13970, 1:1000), mouse anti-armadillo (DSHB, N2 7A1, 1:100), mouse anti-Delta (DSHB, C594.9B, 1:50), mouse anti-Dlg (DSHB, 4F3 anti-discs large, 1:20), mouse anti-ChAT (DSHB, ChAT4B1, 1:100), rabbit anti-phospho Histone H3 (Millipore, 06-570, 1:2000), mouse anti-prospero (DSHB, MR1A, 1:250), mouse anti-cut (DSHB, 2B10, 1:100), mouse anti-Golgin84 (DSHB, Golgin 84 12-1, neat), mouse anti-Coracle heavy isoform (DSHB, C615.16, 1:20).

Secondary antibodies were from Jackson ImmunoResearch Laboratories and diluted 1:1000, donkey anti-mouse Cy3 (Jackson ImmunoResearch Laboratories, 715-165-150, 1:1000), donkey anti-mouse Alexa-647 (Invitrogen, A31571, 1:1000), donkey anti-chicken Alexa-488 (Jackson ImmunoResearch Laboratories, 703-545-155, 1:1000), donkey anti-rabbit Cy3 (Jackson ImmunoResearch Laboratories, 711-165-152, 1:1000), donkey anti-rabbit Alexa-647 (Jackson ImmunoResearch Laboratories, 711-605-152, 1:1000).

All images were taken on a Leica SP8 confocal microscope and processed using FIJI (*Schindelin et al., 2012*) and Adobe Illustrator.

## CFU counting

Commensal bacteria were cultured as described in *Guo et al., 2014*. In brief, intact flies were sanitized in 70% ethanol for 1 min and rinsed 3× in sterile PBS. Five guts per sample were dissected and homogenized in 250 μl sterile PBS. Serial dilutions were plated on selective media, plates were incubated for 48–72 hr at 29°C, and colonies counted.

Selective plates were prepared according to the following recipes: Acetobacteriaceae: 25 g/l D-mannitol, 5 g/l yeast extract, 3 g/l peptone, and 15 g/l agar. Enterobacteriaceae: 10 g/l tryptone, 1.5 g/l yeast extract, 10 g/l glucose, 5 g/l sodium chloride, 12 g/l agar. Lactobacilli MRS agar: 70 g/l BD Difco Lactobacilli MRS agar. Nutrient-rich broth: 23 g/l BD Difco Nutrient agar. All media were autoclaved at 121°C for 15 min.

## Electron microscopy

For the localization of Syt4, flies were allowed to express UAS-3xFLAG-mCherry-Syt4 in ECs under control of NP1ts for 2 days before dissection in PBS at RT. The dissected gut was cut with a sharp blade at the R3-R4 border and the R4-hindgut segment was immediately immersed in either one of two fixative solutions, to obtain samples for immuno-EM and for conventional EM in Epon-embedded material. For immuno-EM (*Slot and Geuze, 2007*), the fixation was performed with 4% paraformaldehyde (PFA), 0.1% glutaraldehyde (GA) in PHEM buffer (60 mM PIPES, 25 mM HEPES, 2 mM MgCl$_2$, 10 mM EGTA), pH 6.9, for 1 hr at RT. Subsequently, fixation was continued in 0.6% PFA in PHEM buffer at 4°C for several days. The samples were then rinsed in PBS, blocked with 0.15% glycine in PBS, and gradually embedded in gelatin, from 2% (30 min) over 6% (30 min) to 12% gelatin. Small blocks of solidified gelatin each containing 1 gut segment were cryoprotected overnight with 2.3 M sucrose. They were mounted on aluminum pins in a direction to expose the transversal cut at the R4 segment for cryo-ultramicrotomy and frozen in liquid N2. Syt4 was localized on ultrathin cryosections with polyclonal rabbit anti-RFP antibody (600-401-379, Rockland). Chitin was localized with biotinylated WGA (B-1025-5, Vector laboratories) followed by polyclonal rabbit anti-biotin antibody (100-4198, Rockland). *Drosophila*-specific rabbit anti-Lamp1 antibody was a gift from Andreas Jenny (*Chaudhry et al., 2022*). Antibodies were detected with protein A conjugated with 15 or 10 nm gold particles in a JEOL JEM-1011 electron microscope.

For conventional EM the fixation was performed in 2.5% GA in 0.1 M Sorensen's phosphate buffer (PB), pH 7.4, for 4 hr at RT, then overnight at 4°C. Subsequently, fixation was continued in 0.6% PFA in 0.1 M PB at 4°C for several days. After rinsing in 0.1 M PB, the guts were postfixed with 1% $OsO_4$ and 1.5 % $K_3[Fe(CN)_6]$ in 0.07 M PB, stained en bloc in aqueous 0.5% uranyl acetate, dehydrated in acetone and embedded in Epon. Transverse sections of the R4 gut segments were stained with uranyl acetate and lead citrate and examined in a JEOL JEM-1011 electron microscope.

## Bulk RNAseq

For bulk RNAseq analysis four independent biological replicates per sample consisting of 20–25 guts each were dissected and collected on dry ice. RNA was extracted using the QIAGEN RNeasy Mini kit.

Total RNA was quantified with Qubit RNA HS Assay Kit (Thermo Fisher Scientific) and quality was assessed using RNA ScreenTape on 4200 TapeStation (Agilent Technologies). For sequencing library generation, the Truseq Stranded mRNA kit (Illumina) was used with an input of 100 ng of total RNA. Libraries were quantified with Qubit dsDNA HS Assay Kit (Thermo Fisher Scientific) and the average library size was determined using D1000 ScreenTape on 4200 TapeStation (Agilent Technologies). Libraries were pooled and sequenced on NovaSeq 6000 (Illumina) to generate 30 millions single-end 50-base pair reads for each sample. Reads were aligned to the *Drosophila* genome, version BDGP6, using the GSNAP aligner as part of the HTSeqGenie R package (version 4.2). Reads that uniquely aligned within exonic boundaries of genes were used to derive expression estimates. nRPKM values, in which total library sizes were normalized using the median ratio method as previously described (*Anders and Huber, 2010*), were generated for each gene. Partek Flow was used to perform differential gene expression and PCA, GO term enrichment as well as creation of illustrative graphs.

## AFCA scRNAseq analysis

The analysis of snRNAseq data relies on the AFCA data set (*Lu et al., 2023*). We specifically focus on the gut cell types, such as ISC, enteroblast, adult differentiating EC, EC, and EE, which were selected from the 5-day AFCA data and further subclustered. Before generating the Uniform Manifold Approximation and Projection plots, we took into account sex and tissue differences by employing the hormony correction method to adjust the principal components (*Korsunsky et al., 2019*).

## RT-qPCR

Three to five independent biological replicates consisting of 20–25 guts per sample were dissected and collected on dry ice. RNA was extracted using the QIAGEN RNeasy Mini kit. 25 ng RNA were used as input for the TaqMan RNA-to-CT 1-Step Kit (Applied Biosciences) in a 384-well format. Assays were run on a QuantStudio 6 real-time PCR system according to the kit instructions and analyzed using the ΔΔCT method (normalized to GAPDH).

Taqman FAM-MGB probes used in this study (Thermo Fisher): Syt4 (Dm02135118_g1), ChAT (Dm02134803_m1), Ace (Dm02134758_g1), nAchR α1(Dm02151345_m1), nAchR α2(Dm02150710_m1), nAchR α3(Dm01843751_m1), nAchR α4(Dm01843901_m1), nAchR α5(Dm01808491_g1), nAchR α6(Dm01803895_m1), nAchR α7(Dm01799687_m1), nAchR β1(Dm01822104_m1), nAchR β2(Dm02150716_g1), nAchR β3(Dm01843796_g1), GADPH (Dm01843827_s1).

## Statistical analyses

Generation of graphs and statistical analyses were performed with GraphPad Prism 9.

Experiments with two conditions were compared with a two-tailed parametric Student's t-test or Fisher's exact test, as appropriate. Experiments with multiple conditions were evaluated either by ordinary one-way ANOVA followed by Dunnett's post hoc test to compare a control group with experimental conditions or a Chi square test for categorical data. Barrier dysfunction curves were analyzed with two-way repeated measures ANOVA. Survival curves were compared with the Mantel-Cox method.

No statistical methods were used to predetermine sample sizes; sample sizes were determined based on variation observed in pilot experiments and previously published literature. Exact numbers are listed in figure legends. All animals were randomly allocated to treatment groups. The experimenter was blinded for image analysis and other quantifications. The number of repeats for each experiment is listed in figure legends, all attempts at replication were successful. The initial screen as

well as electron microscopy and RNAseq experiments were performed once for data gathering and hypothesis generation; the data was later validated by other methods. No data points were excluded from analyses.

### Illustrative model

Illustrative model summarizing results was created with BioRender.com.

### Material availability

Fly lines generated in this study (UAS-ChAT and Syt4 CRISPR mutant) are available without restriction upon an agreement with a material transfer agreement.

### Adherence to community standards

This study and manuscript were prepared in accordance with ARRIVE and ICMJE guidelines.

## Acknowledgements

We thank Dr Andreas Jenny (Albert Einstein College of Medicine) for the *Drosophila*-specific rabbit anti-Lamp1 antibody. We thank Dr Aniek Janssen and Dr Lucie van Leeuwen (UMC Utrecht) for the Oregon R flies for EM.

## Additional information

### Competing interests

Nadja S Katheder, Kristen C Browder, Diana Chang, Zijuan Lai, Dewakar Sangaraju, Heinrich Jasper: is affiliated with Genentech. The author has no financial interests to declare. The other authors declare that no competing interests exist.

### Funding

| Funder | Grant reference number | Author |
| --- | --- | --- |
| Netherlands Organization for Scientific Research | 184.034.014 | Judith Klumperman |

The funders had no role in study design, data collection and interpretation, or the decision to submit the work for publication.

### Author contributions

Nadja S Katheder, Conceptualization, Data curation, Formal analysis, Investigation, Methodology, Writing – original draft, Project administration, Writing – review and editing; Kristen C Browder, Conceptualization, Data curation, Investigation, Methodology; Diana Chang, Conceptualization, Data curation, Formal analysis, Investigation, Methodology; Ann De Maziere, Judith Klumperman, Data curation, Formal analysis, Investigation, Methodology; Pekka Kujala, Suzanne van Dijk, Formal analysis, Investigation, Methodology; Tzu-Chiao Lu, Hongjie Li, Data curation; Zijuan Lai, Data curation, Investigation; Dewakar Sangaraju, Data curation, Formal analysis, Investigation; Heinrich Jasper, Conceptualization, Resources, Data curation, Supervision, Funding acquisition, Writing – review and editing

### Author ORCIDs

Nadja S Katheder https://orcid.org/0000-0003-0460-0938
Ann De Maziere http://orcid.org/0000-0001-8070-5104
Heinrich Jasper https://orcid.org/0000-0002-6014-4343

### Decision letter and Author response

Decision letter https://doi.org/10.7554/eLife.86381.sa1
Author response https://doi.org/10.7554/eLife.86381.sa2

## Additional files

### Supplementary files
- MDAR checklist

- Supplementary file 1. List of candidate genes for genetic variants (human) associated with COPD (*Hobbs et al., 2017*). Genes highlighted in blue had a clear *Drosophila* ortholog and were included in the screen. Abbreviations used: SNP, single nucleotide polymorphism; CHR, chromosome; BP, base pair (GRCh37); eQTL, expression quantitative trait loci; risk allele, allele associated with increased COPD risk; Alt allele, alternative allele; OR stage1, odds ratio of risk allele in stage 1 of P.stage1, p-value in stage 1 of Hobbs et al.; P.meta, meta-analysis p-value in Hobbs et al.; Evidence. Sakornsakolpat, evidence (if available) from *Sakornsakolpat et al., 2019* (GREx-genetically regulated expression, mQTL-methylation quantitative trait loci, Cod-coding association, Hi-C-chromatin interaction in human lung or IMR90 cell line, DHS-DNase hypersensitivity sites, GSet-genes identified by DEPICT, further details are available in the original publication); colocalization, probability shared causal variant between eQTL (GTEx) and COPD risk association (tissue: probability), only colocalization probability > 0.6 are listed.

- Supplementary file 2. List of *Drosophila* genes and RNAi lines included in the screen. RNAi lines were ranked according to the natural logarithm of the ratio between the proportion of smurfs after candidate gene knockdown and Luciferase RNAi control. Cutoff scale shown in *Figure 1C* was used to determine the effect of each RNAi. Based on this fine-grained ranking of individual RNAi lines, an overall rating was assigned to each gene and compared to human eQTL data (see also *Figure 1A*). Temperature column refers to the temperature the subsets of RNAi lines were screened at.

### Data availability
All data generated and analyzed are included in the manuscript, figures and figure supplements. All sequencing data has been deposited in GEO under accession code GSE236071.

The following dataset was generated:

| Author(s) | Year | Dataset title | Dataset URL | Database and Identifier |
|---|---|---|---|---|
| Katheder NS | 2023 | Short-term depletion in *Drosophila* enteroendocrine cells | https://www.ncbi.nlm.nih.gov/geo/query/acc.cgi?acc=GSE236071 | NCBI Gene Expression Omnibus, GSE236071 |

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
