## [Editor Report]

This study presents valuable findings on the maintenance of the intestinal epithelial barrier function in *Drosophila* that involves the production of the peritrophic matrix rather than the epithelial junctions, and highlights a critical role of Ach/nAchR signaling there. The methods, data, and analyses broadly support the claims in line with current state-of-the-art.

---

## [Decision Letter]

**Decision letter after peer review:**

Thank you for submitting your article "Nicotinic acetylcholine receptor signaling maintains epithelial barrier integrity" for consideration by *eLife*. Your article has been reviewed by 3 peer reviewers, and the evaluation has been overseen by a Reviewing Editor and David James as the Senior Editor. The following individual involved in the review of your submission has agreed to reveal their identity: Rongwen Xi (Reviewer #3).

The reviewers have discussed their reviews with one another and find that the manuscript has many merits but still some weaknesses. They ask you to revise the manuscript. The Reviewing Editor has drafted this to help you prepare a revised submission.

Essential revisions:

1. The source of the ligand (Acetylcholine) should better elucidate. Is it produced in enteric neurons and/or EE cells?

The effect on epithelial barrier function resulting from reducing ChaT levels in 'EEs' (prosepro-gal4) versus cholinergic neurons and EEs (ChaT-gal4) is different. The impact is clearly much stronger when the prospero-gal4 driver is used.

Prospero-gal4 would target brain and VNC cells in addition to EE cells. As such, the effects observed under Pros-gal4 driven Ach knockdown may be a consequence of impairing broader ligand action. This needs to be considered and tested experimentally, for example, by targeting neuronally produced Ach while sparing EE cells and seeing how this compares to the effect caused by Prospero-gal4. Alternatively, using a tool to target ChaT from EE cells, while spearing ChaT+ve neurons.

On the other hand, the ChaT-gal4 used may not target all Ach-producing cells. Details of ChAT expression pattern in EEs using antibody staining should be included, by staining ChaT>reporter guts with ChaT antibody. This would also be important considering previous work by Buchon et al. (2013 Cell Rep) showing the ChaT-Gal4 (Bloomington #6793) was expressed in small-nucleated cells in R3. This contrasts with the observation here that ChAT is expressed in EEs in R4-R5 regions indicated by ChAT[MI04508-TG4.0].

2. Is the Ach/AchR pathway regulated upon intestinal damage?

While the data clearly show the constitute role of AchR signaling in the maintenance of intestinal homeostasis (as per the data in Figure 2E-H) and epithelial barrier function following damage exposure (rest of the paper), it is unclear whether pathway activity is regulated by damage.

For instance: is the production of Acetylcholine by EE cells or nerve cells regulated upon intestinal damage? Is AchR activity/signaling in the intestinal epithelium regulated by similar stimuli?

3. Is the effect of AchR knockdown on EE cell differentiation a reflection of an intrinsic role of the signalling in ISCs? Or part of its EC function?

The intestinal cell types that express nAchR genes are not shown. Are they expressed in all intestinal cells including intestinal progenitors (ISC, EB(NotcH^+^), and EMC/EEP), and differentiated EC/EE? Reduced production of EE in nAchRa2 mutant clones suggests a requirement of nAchR at least in the stem cell lineage that produces EE. Following this, do nAchR genes have any role in ISCs for their division and differentiation? Such data are important for assessing the roles of Ach/nAchR axis in regulating intestinal homeostasis.

Do you see defective EE differentiation upon EC-specific knockdown of AChR?

What happens to epithelial barrier function, ISC proliferation, and EE differentiation if AChR is knockdown in stem cells specifically (e.g. using ISC-gal4 drivers or ISC-gene switch system)? These experiments would be important to distinguish potential cell-specific roles of the receptor and likely clarify whether EE cell differentiation and epithelial barrier function are part of the same process and linked to the functional role of the receptor in ECs or not.

Figure 2H (MARM clones): Would you not expect to see bigger clones upon AchR depletion given that its targeted knockdown in ECs by RNAi results in large increase in ISC proliferation reported in (Figure 2E)?

4. The author should better support the functional connection of Ach/AchR signalling and Syt4.

While the data provided shows that the downregulation of Syt4 mimics that of AchR signaling, it does not demonstrate that they are functionally linked. Validation of syt4 expression by qPCR should be performed and assessment of Syt4 protein levels and cellular localization following AchR knockdown. Genetic interaction experiments should be done to confirm that AchR/Syt4 work in the same pathway to regulate PM/barrier integrity. The authors should also test if EC-specific Syt4 depletion and Syt4 mutants show increased susceptibility to paraquat-induced damage as well, although they did show that these flies were less resistant to P.e. infection. Paraquat challenge and P.e. infection likely induce epithelial damage through very different mechanisms. Testing if PM integrity has a role in protecting flies from various stress/damages helps extend the current view of the PM function.

More generally, the paper technically shifted from Smurf assay upon paraquat challenge to survival analysis upon P.e. infection after coming to the PM. Since the susceptibility to Pe infection does not necessarily indicate barrier dysfunction, Smurf assay data should be still included along with the survival curve upon Pe infection. This will not only allow a better assessment of the significance of Ach/nAchR/Syt4 axis in safeguarding epithelial barrier under various conditions (oxidative stress, bacterial infection…) but also keeps the whole paper technically consistent.

5. More careful controls for genetic background and readout of epithelial integrity.

a) About reduced EE generation in alpha2 clones: only one allele was studied, does the RNAi clone or clone mutant for other subunits display a similar phenotype?

b) The authors should also analyze the intercellular junctions that are critical in creating the epithelial barrier function. This should help justify the rationale of their experimental design, even though this paper finally uncovered that nAchR signaling does not seem to regulate barrier function via a junctional mechanism.

c) Several points exist regarding the control used for experiments.

(1) The use of >mCherry-RNAi (at attP2 site) as a control for all RNAi experiments is not appropriate, as this study used many RNAi lines of different genetic backgrounds, including lines from VDRC and also some TRiP lines that may not be inserted in attP2 site.

(2) The new Syt4 crispr mutant should be ideally isogenized to *w1118* which was used as a control in the paper.

Alternatively, a deficiency spanning the syt4 gene can be used to rule out potential background effects associated with the crispr mutant.

(3) For experiments using GeneSwitch, it is ideal to use GS>RNAi flies without RU486 supplementation as control, to make the full potential of the GS system.

---

## [Author Response]

Essential revisions:1. The source of the ligand (Acetylcholine) should better elucidate. Is it produced in enteric neurons and/or EE cells?The effect on epithelial barrier function resulting from reducing ChaT levels in 'EEs' (prosepro-gal4) versus cholinergic neurons and EEs (ChaT-gal4) is different. The impact is clearly much stronger when the prospero-gal4 driver is used.Prospero-gal4 would target brain and VNC cells in addition to EE cells. As such, the effects observed under Pros-gal4 driven Ach knockdown may be a consequence of impairing broader ligand action. This needs to be considered and tested experimentally, for example, by targeting neuronally produced Ach while sparing EE cells and seeing how this compares to the effect caused by Prospero-gal4. Alternatively, using a tool to target ChaT from EE cells, while spearing ChaT+ve neurons.On the other hand, the ChaT-gal4 used may not target all Ach-producing cells. Details of ChAT expression pattern in EEs using antibody staining should be included, by staining ChaT>reporter guts with ChaT antibody. This would also be important considering previous work by Buchon et al. (2013 Cell Rep) showing the ChaT-Gal4 (Bloomington #6793) was expressed in small-nucleated cells in R3. This contrasts with the observation here that ChAT is expressed in EEs in R4-R5 regions indicated by ChAT[MI04508-TG4.0].

We thank the reviewer for this question and we agree that it would be interesting to identify the specific source of acetylcholine (Ach) in specific conditions. We have now performed additional studies as requested (summarized below), but unfortunately have not been able to definitively separate neuronal from enteroendocrine Ach sources due to a lack of specific drivers that only affect one cell type. We are including additional driver characterizations in the manuscript (see below) and have also edited the discussion to reflect the current uncertainty about the most relevant source of the ligand. At this point, the most reasonable interpretation of our data is that Ach derived from both EEs and cholinergic neurons (as reflected by ChAT and Pros expression patterns) can act through Ach receptors on epithelial cells to promote barrier function.

Specifically, we have performed the following additional analyses:

– We are including images showing that pros-Gal4 is also expressed in enteric neurons (Figure 3 —figure supplement 1D), and that the main difference between pros-gal4 and ChAT-Gal4 is that pros-Gal4 expresses in all EEs, while Chat is only expressed in a subset of EEs. It is thus expected that pros-Gal results in a stronger effect when used to manipulate Ach levels.

– As requested, we have now performed antibody staining against ChAT in the ChAT-Gal4>GFP background and found that the antibody labels the ChAT-Gal4+ EEs, as well as additional EEs in R4, suggesting that the driver does not capture all ChAT-expressing cells (Figure 3 —figure supplement 1B).

– ChAT-Gal4 BL6793 is based on a 7.4kb BamHI 5' flanking fragment from the "cholinergic" locus (which encodes both ChAT and VAChT) that regulates expression of Gal4 (Salvaterra and Kitamoto, 2001) The ChAT-Gal4 BL60317 driver (w[*]; Mi{Trojan-GAL4.0}ChAT[MI04508-TG4.0]CG7715[MI04508-TG4.0-X]/TM6B, Tb[1]) in turn, is based on a MIMIC insertion into the ChAT gene. The two drivers were compared (Diao et al., 2015) and it was reported that ChAT-Gal4 7.4 doesn’t label all putative cholinergic pCC interneurons (Figure 1C in that paper), while UAS-ChAT rescues the hatching deficits of a ChaMI04508-Gal4/ChaL13 mutants (table 2). These data suggest that ChAT-Gal4 7.4 does not capture all regulatory sequences, while the MIMIC line, which is inserted into the actual genomic locus, more accurately reflects endogenous gene expression. To further corroborate this, we used BL84618 (TI{2A-GAL4}ChAT[2A-GAL4]/TM3, Sb[1]), which was established independently in a CRISPR KO line followed by phiC31 mediated knock-in of ChAT fused to a T2A-Gal4 sequence inserted at or near the 3' end of ChAT, resulting in expression of Gal4 as a separate protein under the control of ChAT regulation ((Deng et al., 2019),Figure 1B and 1C). This line shows the same pattern as BL60317 (Figure 3 —figure supplement 1A, both drivers label EEs in R2 and R4+R5, but not in R3), suggesting that the R3 expression seen with the artificial promoter construct in ChAT-Gal4 BL6793 is likely not a reflection of endogenous ChAT expression.

– We tested previously reported drivers associated with EE expression, such as BL84614 (w* TI{2A-GAL4}CG325472A-GAL4/FM7a, (Guo et al., 2019)), which also shows expression in enteric neurons (Figure 3 —figure supplement 2A) and BL25373 (dimm-Gal4,(Beebe et al., 2015)), which displayed broad epithelial expression (Figure 3 —figure supplement 2B). Furthermore, we selected candidate drivers from a list of neuropeptides (Chen et al., 2016) with low expression in the brain based on scRNAseq data (Davie et al., 2018), but all tested drivers were either not specific to EEs, expressed in other epithelial cells or showed neuronal expression (CCAP-Gal4, Burs-Gal4, Dsk-Gal4, Nplp4-Gal4, Figure 3 —figure supplement 2B).

– We used Fly Cell Atlas scRNAseq data (Li et al., 2022) to identify additional EE drivers with low neuronal expression. Out of 3 candidate genes, only one had a Gal4 driver available (BL92253, y1 w*; TI{CRIMIC.TG4.1}OrcokininCR02407-TG4.1, Orcokinin-Gal4). While displaying low enteric neuron expression, Orcokinin-Gal4 is unfortunately not entirely EE-specific and labels ECs as well, which led us to not consider this driver further (Figure 3 —figure supplement 2A).

– We pursued a split-gal4 approach using a EE-reference driver (BL91402, w[*]; P{y[+t7.7] w[+mC]=R57F07-p65.AD.A}attP40; P{y[+t7.7] w[+mC]=UAS-DSCP-6XEGFP}attP2) tested by (Holsopple et al., 2022) and crossed it to a small set of DBD split gal4 drivers (Figure 3 —figure supplement 3). First, we crossed the reference driver to its own DBD counterpart (BL91403), which led to neuronal expression (Figure 3 —figure supplement 3A). We then identified two DBD drivers (BL68537 and BL69158) that were expressed in all gut compartments but not in the brain, using the table available at https://bdsc.indiana.edu/stocks/gal4/midgut_EEs.html. While the BL91402/Bl68537 pair was expressed in enteric neurons, the BL91402/BL69158 combination showed no expression in neurons, but only labeled a very small subset of EEs (Figure 3 —figure supplement 3B). When crossed to a ChAT-DBD driver, we were unable to detect any GFP expression in the epithelium (Figure 3 —figure supplement 3C). Additional characterization of split Gal4 combinations may be a potential avenue to separate expression in EEs and neurons, but extensive screening work will have to be performed. We believe that this would be beyond the scope of the current manuscript.

2. Is the Ach/AchR pathway regulated upon intestinal damage?While the data clearly show the constitute role of AchR signaling in the maintenance of intestinal homeostasis (as per the data in Figure 2E-H) and epithelial barrier function following damage exposure (rest of the paper), it is unclear whether pathway activity is regulated by damage.For instance: is the production of Acetylcholine by EE cells or nerve cells regulated upon intestinal damage? Is AchR activity/signaling in the intestinal epithelium regulated by similar stimuli?

We agree that this is a very interesting question and have now included several experiments that explore this:

– We have quantified the number of ChAT>GFP-positive EEs after *Pseudomonas* entomophila (PE) infection, but saw no increase (Figure 5 —figure supplement 1C).

– We have performed RT-qPCR on whole guts after PE infection, and found an induction of transcripts for nAchRβ3 (but not ChAT or other nAchR subunits) as well as a decrease of acetylcholine esterase (Ace) transcript levels (Figure 5 —figure supplement 1B), which could suggest a stabilization/increase of Ach levels after injury.

– We’ve also attempted to measure Ach concentration in the whole fly after infection biochemically, but unfortunately results were inconclusive due to difficulties in sample collection (large amounts of tissue needs to be collected in a very short period of time to overcome the detection limit of the assay while preventing Ach degradation).

While induction of nAchRβ3 and the suppression of Ace suggest possible activation of the pathway after infection, further studies with high-speed live imaging of Ach reporters would be needed to definitively answer the question of how and where Ach signaling is regulated in injured intestines. We believe that properly establishing such techniques and performing such studies would constitute a very interesting but exhaustive follow-up study that is beyond the scope of the current study.

3. Is the effect of AchR knockdown on EE cell differentiation a reflection of an intrinsic role of the signalling in ISCs? Or part of its EC function?The intestinal cell types that express nAchR genes are not shown. Are they expressed in all intestinal cells including intestinal progenitors (ISC, EB(NotcH^+^), and EMC/EEP), and differentiated EC/EE? Reduced production of EE in nAchRa2 mutant clones suggests a requirement of nAchR at least in the stem cell lineage that produces EE. Following this, do nAchR genes have any role in ISCs for their division and differentiation? Such data are important for assessing the roles of Ach/nAchR axis in regulating intestinal homeostasis.Do you see defective EE differentiation upon EC-specific knockdown of AChR?What happens to epithelial barrier function, ISC proliferation, and EE differentiation if AChR is knockdown in stem cells specifically (e.g. using ISC-gal4 drivers or ISC-gene switch system)? These experiments would be important to distinguish potential cell-specific roles of the receptor and likely clarify whether EE cell differentiation and epithelial barrier function are part of the same process and linked to the functional role of the receptor in ECs or not.

We agree that this is a very interesting question, and we have now performed various additional studies to explore the role of nAchR signaling in ISC to EE differentiation:

– To confirm our nAchR α2 MARCM data, we utilized the esgF/O strategy (Jiang et al., 2009) to knock down the expression of other subunits in clones via RNAi and observed a mild trend towards fewer EEs in α2 and α4 knockdowns (Figure 2 —figure supplement 2F). We also made use of a null allele for the α1 subunit of nAchR to generate MARCM clones and once more observed a trend towards fewer EEs per clone (Figure 2 —figure supplement 2E). Since the effects of these perturbations on EE numbers were mild, we have edited our description of the impact of nAchR signaling on EE differentiation. Our data suggest that there is no absolute requirement for nAchR signaling in EE differentiation, but that the robustness of lineage decisions are impacted by Ach.

– Consistent with the limited impact on clone sizes in the MARCM or F/O experiments, knockdown of various nAchR subunits with an intestinal stem cell (ISC) specific-driver (esg-Gal4, UAS-2xEYFP; Su(H)GBE-Gal80, tubGal80ts) for 72h did not impact ISC proliferation after PE infection (Figure 2 —figure supplement 2D), while still increasing barrier dysfunction and causing reduced survival after challenge with either paraquat or *Pseudomonas* entomophila (PE) (Figure 2 —figure supplement 2B,C). Together, these observations suggest that the barrier dysfunction and mortality may be caused by changes in ISC daughter cells rather than by changes in ISC proliferative capacity. It should be noted that the increase in pH3+ cells we report after EB/EC knockdown of nAchR subunits in Figure 2E is the expected proliferation response of ISCs to stressed ECs that secrete Upd cytokines (Jiang et al., 2009) and is thus uncoupled from the effects of nAchR knockdown in ISCs selectively. We have edited the discussion to clarify this point.

– Interestingly, when nAchR subunits were knocked down specifically in EBs and ECs for 8 days, we observed no effect on the number of EEs (Figure 3 —figure supplement 1C), suggesting that nAchR may be influencing the lineage commitment of ISCs to the EE fate prior to EB formation.

– To determine the cell-specific expression pattern of nAchR signaling components across the gut, we referenced scRNAseq data recently reported by us in the Aging Fly Cell Atlas ((Lu et al., 2023), Figure 4 —figure supplement 2). Overall, all nAchR subunits as well as the Ach-producing enzyme ChAT are lowly expressed in the gut epithelium of 5 day old animals, with α4 showing the highest expression levels across all cell types. All subunits are expressed in ECs and EEs, while expression in ISCs and EBs is more variable. Subunit α5 shows an enrichment in EEs. ChAT is preferentially expressed in EEs, but also shows some residual EC expression. The Ach-degrading enzyme Ace on the other hand is more widely expressed and shows an enrichment in ECs, potentially suggesting a local modulation of Ach levels in the gut epithelium. We are now discussing these observations in the manuscript (Figure 4 —figure supplement 2).

Figure 2H (MARM clones): Would you not expect to see bigger clones upon AchR depletion given that its targeted knockdown in ECs by RNAi results in large increase in ISC proliferation reported in (Figure 2E)?

Yes, we were a bit surprised by this phenotype, but we believe that it is a reflection of cell-autonomous vs non-autonomous effects of nAchR perturbations as discussed above. In the MARCM experiments only newly formed ECs (and not existing ECs) will be deficient in nAchR, and thus the majority of EC secreted factors (including the peritrophic matrix) will still be intact. When we knock down nAchR in all ECs at once, the perturbation will affect the whole gut and thus result in stress-induced ISC proliferation.

4. The author should better support the functional connection of Ach/AchR signalling and Syt4.While the data provided shows that the downregulation of Syt4 mimics that of AchR signaling, it does not demonstrate that they are functionally linked. Validation of syt4 expression by qPCR should be performed and assessment of Syt4 protein levels and cellular localization following AchR knockdown. Genetic interaction experiments should be done to confirm that AchR/Syt4 work in the same pathway to regulate PM/barrier integrity. The authors should also test if EC-specific Syt4 depletion and Syt4 mutants show increased susceptibility to paraquat-induced damage as well, although they did show that these flies were less resistant to P.e. infection. Paraquat challenge and P.e. infection likely induce epithelial damage through very different mechanisms. Testing if PM integrity has a role in protecting flies from various stress/damages helps extend the current view of the PM function

We agree with the need to further explore these interactions, and have now performed additional studies to shed light on the role of Syt4:

– We confirmed a reduction of Syt4 transcript levels with RT-qPCR after nAchR knockdown in ECs (Figure 4 —figure supplement 1B).

– We performed genetic interaction experiments as suggested: Combined knockdown of nAchR α4 and Syt4 did not worsen sensitivity to PE infection (Figure S9E) compared to individual knockdown of both genes, indicating that they act in the same pathway. While overexpression of 3xFLAG-mCherry-Syt4 was protective against PE infection (Figure 6E), we were unable to rescue PE infection sensitivity when knocking down AchR and overexpressing Syt4 concomitantly, suggesting that Syt4 expression is required but not sufficient for epithelial integrity downstream of nAchR signaling (Figure 6 —figure supplement 1I,I’,I’’).

– We have also performed the proposed studies to explore the effect of Paraquat on barrier dysfunction in the Syt4 loss of function conditions. Flies lacking Syt4 are more sensitive to PE infection and display barrier dysfunction (Figure 6C, Figure 6 —figure supplement 1F), but do not display an enhanced sensitivity to paraquat (Figure 6 —figure supplement 1G). Knockdown of Syt4 in ECs leads to inconclusive sensitivity to Paraquat, with some experiments showing elevated sensitivity, and other showing very limited effects (Figure 6 —figure supplement 1H). This surprising observation is in line with findings reported by Kuraishi and colleagues (Kuraishi et al., 2011): Dcy-deficient flies show PM defects and are very sensitive to bacterial infection, but do not succumb to Paraquat challenge, which is attributed to a role for the PM in protecting the animal against bacterial toxins. This further supports our interpretation that maintenance of PM integrity is an important aspect contributing to barrier integrity downstream of nAchR signaling.

– For localization studies, we tested 2 commercial antibodies against human Syt4 (HPA010574 (Σ) and PA5-52709 (Invitrogen)), but with both antibodies we did not obtain consistent, Syt4-dependent staining. We were unfortunately not able to procure a previously reported antibody against *Drosophila* Syt4 (Adolfsen et al., 2004) or to rapidly make our own antibody.

More generally, the paper technically shifted from Smurf assay upon paraquat challenge to survival analysis upon P.e. infection after coming to the PM. Since the susceptibility to Pe infection does not necessarily indicate barrier dysfunction, Smurf assay data should be still included along with the survival curve upon Pe infection. This will not only allow a better assessment of the significance of Ach/nAchR/Syt4 axis in safeguarding epithelial barrier under various conditions (oxidative stress, bacterial infection…) but also keeps the whole paper technically consistent.

– To address the reviewer’s concern, we coupled PE infection with the smurf assay by resuspending the bacteria in dye-containing sucrose solution. We observed a clear increase of barrier dysfunction in Syt4-deficient animals compared to *w1118* controls (Figure 6 —figure supplement 1F), pointing to the importance of the PM in protecting the animal against pathogenic virulence factors.

5. More careful controls for genetic background and readout of epithelial integrity.a) About reduced EE generation in alpha2 clones: only one allele was studied, does the RNAi clone or clone mutant for other subunits display a similar phenotype?

As discussed in our response to point 3, we have performed additional studies knocking down nAchR subunits and inducing MARCM clones using an α1 loss of function mutation (Figure 2 —figure supplement 2E, F). In both cases, there was a weak trend towards reduced EE differentiation. We don’t believe these effects on EE differentiation are critical for other phenotypes, as they only show a partial reduction in EE numbers, and we have now modified the discussion to clarify this point.

b) The authors should also analyze the intercellular junctions that are critical in creating the epithelial barrier function. This should help justify the rationale of their experimental design, even though this paper finally uncovered that nAchR signaling does not seem to regulate barrier function via a junctional mechanism.

We have now performed stainings for the septate junction markers Dlg and coracle, and focused on cross sections of the epithelium to show that the localization of both proteins is mostly unchanged after nAchR knockdown (Figure 2 —figure supplement 2A). Additionally, we assessed the localization of GFP-tagged armadillo protein, which was not affected by nAchR α2 knockdown. (Figure 2I).

c) Several points exist regarding the control used for experiments.1) The use of >mCherry-RNAi (at attP2 site) as a control for all RNAi experiments is not appropriate, as this study used many RNAi lines of different genetic backgrounds, including lines from VDRC and also some TRiP lines that may not be inserted in attP2 site.

We agree with the reviewer that choice of appropriate controls is important and have now performed a range of control crosses with NP1-Gal4, tubGal80^ts^ to assess the contribution of different backgrounds to barrier dysfunction using the smurf assay (Figure 2 —figure supplement 1A). The lines included are:

– *w1118* as recommended by VDRC as a control for the GD collection, in which inverted repeats are inserted by P-element insertion.

– BL35785: Expresses dsRNA for RNAi of mCherry under UAS control in the VALIUM20 vector; attP2.

– BL31603: Expresses dsRNA for RNAi of Ppyr\LUC (FBgn0014448) under UAS control in the VALIUM1 vector; attP2.

– BL35789: Expresses firefly Luciferase under the control of UAS in the VALIUM1 vector; attP2.

– BL35788: Expresses firefly Luciferase under the control of UAS in the VALIUM10 vector.

– Can be used as a control for VALIUM10 or VALIUM20; attP2.

– BL35786: Expresses GFP under the control of UAS in the VALIUM10 vector.

– Can be used as a control for VALIUM10 or VALIUM20; attP2.

– BL36304: 2nd chromosome attP docking site for phiC31 integrase-mediated transformation.

– Note that the attP40 docking site is located within Msp300 and it may disrupt gene function.

– BL36303: 3rd chromosome attP docking site for phiC31 integrase-mediated transformation.

– v11392 nAchR α4 RNAi as a positive control.

We did not detect any significant differences between the control lines.

2) The new Syt4 crispr mutant should be ideally isogenized to w1118 which was used as a control in the paper.Alternatively, a deficiency spanning the syt4 gene can be used to rule out potential background effects associated with the crispr mutant.3) For experiments using GeneSwitch, it is ideal to use GS>RNAi flies without RU486 supplementation as control, to make the full potential of the GS system.

We thank the reviewer for bringing up this point and have now used a deficiency covering the Syt4 locus to address potential background effects. Transheterozygous animals carrying the deficiency and the Syt4 CRISPR allele are equally sensitive to PE infection as homozygous Syt4 CRISPR mutant animals (Figure 6C, Figure 6 —figure supplement 1F).

For the screen reported in Figure 1, including the -RU control was not practical, but we have confirmed critical phenotypes using NP1-Gal, tubGa80ts as an alternative and independent inducible system.

References:

Adolfsen, B., Saraswati, S., Yoshihara, M., and Littleton, J. T. (2004). Synaptotagmins are trafficked to distinct subcellular domains including the postsynaptic compartment. *J Cell Biol*, *166*(2), 249-260. https://doi.org/10.1083/jcb.200312054

Beebe, K., Park, D., Taghert, P. H., and Micchelli, C. A. (2015). The *Drosophila* Prosecretory Transcription Factor dimmed Is Dynamically Regulated in Adult Enteroendocrine Cells and Protects Against Gram-Negative Infection. *G3 (Bethesda)*, *5*(7), 1517-1524. https://doi.org/10.1534/g3.115.019117

Chen, J., Kim, S. M., and Kwon, J. Y. (2016). A Systematic Analysis of *Drosophila* Regulatory Peptide Expression in Enteroendocrine Cells. *Mol Cells*, *39*(4), 358-366. https://doi.org/10.14348/molcells.2016.0014

Davie, K., Janssens, J., Koldere, D., De Waegeneer, M., Pech, U., Kreft, Ł., Aibar, S., Makhzami, S., Christiaens, V., Bravo González-Blas, C., Poovathingal, S., Hulselmans, G., Spanier, K. I., Moerman, T., Vanspauwen, B., Geurs, S., Voet, T., Lammertyn, J., Thienpont, B.,... Aerts, S. (2018). A Single-Cell Transcriptome Atlas of the Aging *Drosophila* Brain. *Cell*, *174*(4), 982-998.e920. https://doi.org/10.1016/j.cell.2018.05.057

Deng, B., Li, Q., Liu, X., Cao, Y., Li, B., Qian, Y., Xu, R., Mao, R., Zhou, E., Zhang, W., Huang, J., and Rao, Y. (2019). Chemoconnectomics: Mapping Chemical Transmission in *Drosophila*. *Neuron*, *101*(5), 876-893 e874. https://doi.org/10.1016/j.neuron.2019.01.045

Diao, F., Ironfield, H., Luan, H., Diao, F., Shropshire, W. C., Ewer, J., Marr, E., Potter, C. J., Landgraf, M., and White, B. H. (2015). Plug-and-play genetic access to *Drosophila* cell types using exchangeable exon cassettes. *Cell Rep*, *10*(8), 1410-1421. https://doi.org/10.1016/j.celrep.2015.01.059

Guo, X., Yin, C., Yang, F., Zhang, Y., Huang, H., Wang, J., Deng, B., Cai, T., Rao, Y., and Xi, R. (2019). The Cellular Diversity and Transcription Factor Code of *Drosophila* Enteroendocrine Cells. *Cell Rep*, *29*(12), 4172-4185 e4175. https://doi.org/10.1016/j.celrep.2019.11.048

Holsopple, J. M., Cook, K. R., and Popodi, E. M. (2022). Identification of novel split-GAL4 drivers for the characterization of enteroendocrine cells in the *Drosophila melanogaster* midgut. *G3 (Bethesda)*, *12*(6). https://doi.org/10.1093/g3journal/jkac102

Jiang, H., Patel, P. H., Kohlmaier, A., Grenley, M. O., McEwen, D. G., and Edgar, B. A. (2009). Cytokine/Jak/Stat signaling mediates regeneration and homeostasis in the *Drosophila* midgut. *Cell*, *137*(7), 1343-1355. https://doi.org/10.1016/j.cell.2009.05.014

Kummer, W., Lips, K. S., and Pfeil, U. (2008). The epithelial cholinergic system of the airways. *Histochem Cell Biol*, *130*(2), 219-234. https://doi.org/10.1007/s00418-008-0455-2

Kuraishi, T., Binggeli, O., Opota, O., Buchon, N., and Lemaitre, B. (2011). Genetic evidence for a protective role of the peritrophic matrix against intestinal bacterial infection in *Drosophila melanogaster*. *Proc Natl Acad Sci U S A*, *108*(38), 15966-15971. https://doi.org/10.1073/pnas.1105994108

Li, H., Janssens, J., De Waegeneer, M., Kolluru, S. S., Davie, K., Gardeux, V., Saelens, W., David, F. P. A., Brbic, M., Spanier, K., Leskovec, J., McLaughlin, C. N., Xie, Q., Jones, R. C., Brueckner, K., Shim, J., Tattikota, S. G., Schnorrer, F., Rust, K.,... Zinzen, R. P. (2022). Fly Cell Atlas: A single-nucleus transcriptomic atlas of the adult fruit fly. *Science*, *375*(6584), eabk2432. https://doi.org/10.1126/science.abk2432

Lu, T. C., Brbic, M., Park, Y. J., Jackson, T., Chen, J., Kolluru, S. S., Qi, Y., Katheder, N. S., Cai, X. T., Lee, S., Chen, Y. C., Auld, N., Liang, C. Y., Ding, S. H., Welsch, D., D'Souza, S., Pisco, A. O., Jones, R. C., Leskovec, J.,... Li, H. (2023). Aging Fly Cell Atlas identifies exhaustive aging features at cellular resolution. *Science*, *380*(6650), eadg0934. https://doi.org/10.1126/science.adg0934

Salvaterra, P. M., and Kitamoto, T. (2001). *Drosophila* cholinergic neurons and processes visualized with Gal4/UAS-GFP. *Brain Res Gene Expr Patterns*, *1*(1), 73-82. https://doi.org/10.1016/s1567-133x(01)00011-4

Zeng, X., and Hou, S. X. (2015). Enteroendocrine cells are generated from stem cells through a distinct progenitor in the adult *Drosophila* posterior midgut. *Development*, *142*(4), 644-653. https://doi.org/10.1242/dev.113357